# Prism: A Framework for Decoupling and Assessing the Capabilities of VLMs

**Yuxuan Qiao**[2*], **Haodong Duan**[1†], **Xinyu Fang**[4], **Junming Yang**[5], **Lin Chen**[6],
**Songyang Zhang**[1], **Jiaqi Wang**[1], **Dahua Lin**[1,3,7], **Kai Chen**[1†]

[1]Shanghai AI Laboratory    [2]Nanjing University    [3]The Chinese University of Hong Kong
[4]Tongji University    [5]Nanjing University of Posts and Telecommunications
[6]University of Science and Technology of China    [7] CPII under InnoHK
yuxuanqiao@smail.nju.edu.cn
duanhaodong@pjlab.org.cn

## Abstract

Vision Language Models (VLMs) demonstrate remarkable proficiency in addressing a wide array of visual questions, which requires strong perception and reasoning faculties. Assessing these two competencies independently is crucial for model refinement, despite the inherent difficulty due to the intertwined nature of seeing and reasoning in existing VLMs. To tackle this issue, we present **Prism**, an innovative framework designed to disentangle the perception and reasoning processes involved in visual question solving. Prism comprises two distinct stages: a **perception stage** that utilizes a VLM to extract and articulate visual information in textual form, and a **reasoning stage** that formulates responses based on the extracted visual information using a Large Language Model (LLM). This modular design enables the systematic comparison and assessment of both proprietary and open-source VLM for their perception and reasoning strengths. Our analytical framework provides several valuable insights, underscoring Prism's potential as a cost-effective solution for vision-language tasks. By combining a streamlined VLM focused on perception with a powerful LLM tailored for reasoning, Prism achieves superior results in general vision-language tasks while substantially cutting down on training and operational expenses. Quantitative evaluations show that Prism, when configured with a vanilla 2B LLaVA and freely accessible GPT-3.5, delivers performance on par with VLMs $10\times$ larger on the rigorous multimodal benchmark MMStar.

## 1  Introduction

With the rapid development of Large Language Models (LLMs) [45, 59, 60, 56, 5, 19, 4], Vision Language Models (VLMs) [46, 55, 6, 36, 16, 18] have also experienced significant advancements. As an end-to-end approach, VLMs trained on large-scale multimodal data [36, 11, 51, 8] exhibit superior performance on a variety of tasks. These tasks range from basic ones such as object localization [31] and optical character recognition [44, 38] to more complex challenges like document or diagram comprehension [26, 43, 42] and solving geometric problems [40]. For VLMs to solve a general visual question, two essential capabilities are required: 1) **Perception**: extracting necessary information from the image; 2) **Reasoning**: generating answers based on the extracted information and contextual understanding. Limitations in either capability can impede the overall performance of a VLM.

A systematic evaluation of the perception and reasoning capabilities is crucial to provide valuable insights for future model optimization. However, seeing and reasoning are mostly **entangled** in

---

[*] The work was done during an internship at Shanghai AI Laboratory.

[†] Corresponding Author.

38th Conference on Neural Information Processing Systems (NeurIPS 2024).

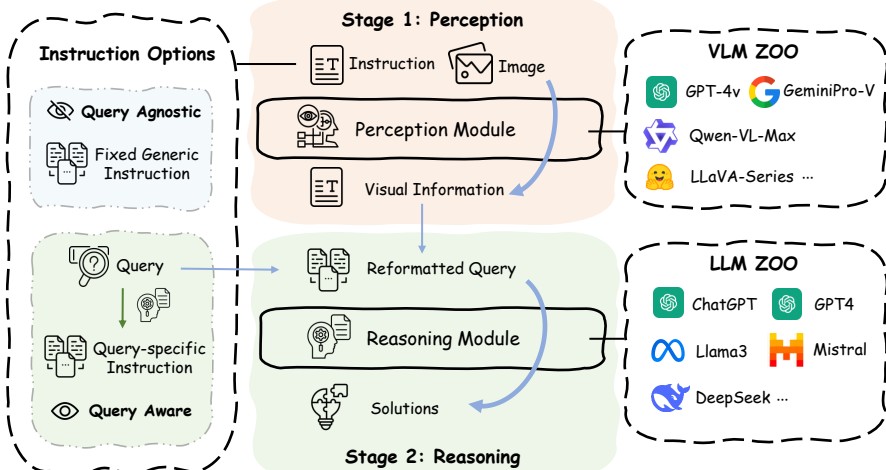

Figure 1: **Prism Framework Architecture.** Prism framework takes image-query pairs as input. An instruction (can be query-agnostic or query-aware) and the image are first fed into the VLM to extract visual information. Then, an LLM is used to generate the answer based on the reformatted query which combines the original question and visual information in textual form.

existing VLMs. Proprietary VLMs, such as GPT-4v are opaque systems that only provide the final answer to visual questions. Meanwhile, open-source VLMs [36] commonly utilize vision encoders to extract visual embeddings, which are often difficult to interpret, from the image and employ adapted LLMs to generate answers based on these visual and linguistic embeddings. In light of this challenge, we introduce **Prism**, a framework designed to disentangle the perception and reasoning processes of any given VLM. It can serve as a proxy for assessing the real perception capabilities of VLMs.

Prism decomposes the process of solving general visual questions into two distinct stages: 1) a **perception stage** that concentrates on extracting visual information from the image using VLMs and articulating this information in textual form; and 2) a **reasoning stage** that utilizes LLMs to answer the question based on the extracted visual information. Prism facilitates the analysis of VLMs' capabilities through two approaches. To assess the true perception capabilities, one can use a constant LLM in the reasoning stage while testing various VLMs in the perception stage. Conversely, by keeping the VLM fixed and varying the LLM in the reasoning stage, one can determine whether a VLM's performance is limited by its reasoning capabilities. Through this analysis, we have uncovered several key insights: 1) Proprietary VLMs, such as GPT-4o and GPT-4v, take the lead in the perception capabilities competition; 2) For open-source VLMs, perception capabilities remain relatively consistent regardless of the language model's size; and 3) The overall performance of open-source VLMs, particularly those with smaller-scale language models like 7B variants, is often constrained by the limited reasoning capabilities.

Beyond its role as an evaluation framework, Prism also excels as an efficient general Vision-Language Model (VLM). Building upon findings 2 and 3, we posit that integrating a small-scale VLM as a visual captioner with a powerful LLM as a reasoning engine offers a promising and efficient strategy for general vision-language tasks. By concentrating on visual information extraction, a lightweight VLM can achieve decent performance on par with much larger VLMs. When paired with a powerful yet economical LLM, thanks to advancements in deployment techniques, one can achieve a robust solution for visual-language information processing, requiring significantly fewer hardware resources for training and deployment. In our experiments, we trained an approximately 2B-parameter vanilla LLaVA to extract visual information and observed that it exhibits perception performance comparable to LLaVA-NeXT [35] that is equipped with a 34B powerful language model. Quantitative evaluations indicate that Prism, when instantiated with a streamlined visual captioner and the freely available ChatGPT-3.5, outperforms many open-source VLMs on multiple multimodal benchmarks including the stringent multimodal understanding benchmark MMStar [10]. Notably, the advantage is particularly pronounced on reasoning-related visual questions.

In summary, the contributions of this work are as follows:

1. We introduce **Prism**, a highly adaptable framework designed to explicitly disentangle the perception and reasoning processes. Prism enables the breakdown analysis of VLM capabilities and serves as a solution for vision-language tasks by integrating any given VLM and LLM.

2. Utilizing Prism, we conduct a **decoupled analysis** of the perception and reasoning capabilities of existing VLMs. Several intriguing findings emerge from the ablation study.

3. Drawing inspiration from these findings, we integrate a lightweight VLM focused on perception with a powerful LLM dedicated to reasoning within the Prism framework. Quantitative results demonstrate that this combination exhibits **outstanding performance and efficiency** across a range of vision-language tasks.

## 2 Methodology

### 2.1 Prism Architecture

Prism is characterized by a modular design that decomposes the process of solving visual questions into two stages, consisting of a perception module and a reasoning module, both of which can be flexibly replaced, as depicted in Fig. 1. The perception module, typically a VLM, initially follows the instruction to extract visual information from images and articulates this information in textual form. The instruction can be generic or query specific, *ie.*, written by the reasoning module given the question (text-only) as contextual information. Meanwhile, the reasoning module, usually an LLM, performs text-based reasoning on the textual information to generate answers to the questions. To assess the perception capabilities of various VLMs, we carefully select an appropriate benchmark corresponding to the principle we illustrate in Sec. 2.2. In Sec. 2.3, we describe how we utilize the Prism framework to assess the perception and reasoning capabilities of VLMs, respectively. To substantiate our belief in the potential of combining a small-scale VLM with a powerful LLM, we train an approximately 2B-parameter VLM based on the LLaVA architecture to serve as a visual captioner and integrate it into Prism as the perceptual module, as detailed in Sec. 2.4.

### 2.2 To Analyze with a Suitable Benchmark

Numerous multimodal benchmarks [37, 20, 28, 40, 66, 10] exist, each evaluating the capabilities of VLMs from various perspectives. To maximize the utility of the Prism framework, careful selection of the benchmark is imperative for the decoupling analysis. In summary, we adhere to the following principles for benchmark selection: 1) **Vision Indispensability**: The benchmark must require visual information for question solving, and questions that can be answered without utilizing visual information are excluded; 2) **Minimal Data Leakage**: The visual questions should not be part of the model's training data; and 3) **Complexity**: The solving process of the visual question should involve both perception and reasoning components. Considering these three principals, we select MMStar [10] as the primary benchmark for the decoupling analysis and ablation. MMStar ensures vision indispensability and makes a concerted effort to minimize data leakage, while many other benchmarks are plagued by the two issues.

### 2.3 Prism as an Analytical Framework

Prism functions as an analytical framework for evaluating the perception and reasoning capabilities of a given VLM. In this section, we elaborate on the methodology for evaluating these capabilities.

To assess the perception capabilities of various VLMs, we employ ChatGPT (GPT-3.5-turbo-0125) as the reasoning module and standardize instructions for description within Prism. With the reasoning module and instructions fixed, the final accuracy of the visual questions is solely determined by the quality of the visual information extracted. Under the controlled setting, we consider the VQA accuracy as a proxy to measure the perception capability of VLMs.

Within the Prism framework, the instructions for visual information extraction are crucial as they are designed to elicit the fundamental perceptual capabilities of VLMs. We have adopted two types of instructions to assess the perceptual abilities of models:

1. **Generic Instruction**: a standardized, universal instruction aimed at extracting and describing the basic elements present in an image;

2. **Query-Specific Instruction**: a combination of the generic instruction and an incremental instruction that directs the VLM to provide a detailed account of the visual information relevant to the question. The incremental instruction is crafted by the reasoning module given the text-only question.

Employed with the generic instruction, Prism offers a straightforward decoupling pipeline. Meanwhile, using query-specific instructions is a more realistic setting and can better realize the full potential of the Prism framework. A Prism decoupled GPT-4o (GPT-4o for both perception and reasoning) achieves almost the same quantitative performance compared to the end-to-end GPT-4o. Analyzing the results, we find that with the reasoning module and instruction controlled, VLMs with language models of different sizes display a much narrowed gap in perception performance compared to the results under the end-to-end setting.

Besides evaluating the perception performance, Prism can also roughly measure the reasoning capabilities of VLMs. When used to solve general visual questions in an end-to-end manner, the VLM implicitly performs reasoning with its language encoder. Alternatively, Prism provides an explicit pipeline, in which an off-the-shelf LLM is separately used to predict the answer based on the visual information. For small-scale VLMs (7B-parameter, *etc.*), we find that Prism equipped with the VLM and ChatGPT-3.5 outperforms the end-to-end VLM in quantitative performance, especially for reasoning related VQA. The results reveal that for small-scale VLMs, the overall performance can be heavily constrained by the parameter size of the language model.

## 2.4 Prism as a Vision-Language Task Solver

Beyond serving as an evaluation framework, Prism can also function as an **efficient vision-language task solver**. The perception module in Prism can incorporate one or multiple VLMs to extract high-quality visual information. Concurrently, the reasoning module can be instantiated with a powerful LLM to harness its advanced reasoning capabilities.

In Sec. 2.3, we observed that VLMs paired with language models of varying sizes exhibit similar performance in perception capability. This suggests that it is promising to employ small-scale VLMs to generate informative visual descriptions, serving as an efficient perception module. To validate this concept, we conduct extensive experiments to train visual captioners, utilizing the widely adopted LLaVA architecture and only open-source datasets. Below, we elaborate on the specific settings considered for this ablation study in detail:

**Instruction Tuning Data.** We use `ALLaVA-Caption-4V` and `Evol-Intruct-GPT4-Turbo-143K` in `ALLaVA` [9] as our instruction tuning data. The former comprises 715K pairs of images and detailed visual captions produced by GPT-4v, whereas the latter contains 143K text instruction tuning data, generated by GPT-4-Turbo. Utilizing the descriptive data for instruction tuning better triggers the VLM's ability to extract and articulate visual information more effectively, compared to instruction tuning data in QA formats [36, 32, 52, 42].

**Model Architecture.** To investigate the impact of vision encoder in LLaVA, we experimented with multiple encoders, including `Clip ViT-L/14` [50], `SigLip-SO400M` [67], and `InternViT-6B` [13]. For the language encoder, we tested two lightweight variants of InternLM2 [7]: `InternLM2-7B` and `InternLM2-1.8B`. The inference of all combinations (excluding `InternViT-6B`) can be efficiently executed on consumer-level GPUs such as RTX 4090, *etc.* During instruction tuning, we maintain the vision encoder fixed and apply QLoRA [17] to the language encoder.

Regarding the vision backbone, `SigLip` exhibits relatively superior performance. Additionally, we observed that a larger language model only results in minor differences in perception performance. Our findings indicate that, within the Prism framework, a 2B vision captioner can achieve strong perception performance on par with LLaVA-NeXT [35] equipped with a 34B language backbone.

## 3 Evaluation Results

### 3.1 Implementation Details

**Evalution Details.** We use Prism to evaluate the capabilities of various VLMs, which can be categorized into two major groups: (a) **Proprietary VLMs**, including GPT-4o (20240513) [46], GPT-4v (20231106) [46], GeminiPro-V [55], and Qwen-VL-Max [6]; (b) **Open-Source VLMs**, including LLaVA-v1.5 [34], InternLM-XComposer2 [18], mPLUG-Owl2 [65], LLaVA-NeXT [35], InternVL-Chat-v1.5 [12], DeepSeek-VL [39], MiniCPM-V-2 [47]. When integrating these VLMs as perception modules within Prism, we employ greedy decoding and limit the maximum number of output tokens to 512. The evaluation encompasses both generic and query-specific instructions.

| Model | Generic Instruction | | | | | | | Query-Specific Instruction | | | | | | |
|---|---|---|---|---|---|---|---|---|---|---|---|---|---|---|
| | CP | FP | IR | LR | Math | ST | Overall | CP | FP | IR | LR | Math | ST | Overall |
| *Proprietary VLMs* | | | | | | | | | | | | | | |
| **GPT-4o [46]** | 64.0 | **41.6** | 54.4 | **51.6** | **47.6** | 31.2 | 48.4 | 67.2 | 51.6 | 63.2 | 56.4 | 52.0 | 36.8 | 54.5 |
| **GPT-4v [46]** | 62.4 | 33.2 | 51.2 | 39.2 | 44.4 | **32.8** | 43.9 | 67.2 | 43.6 | 56.8 | 46.8 | 42.0 | 30.8 | 47.9 |
| **GeminiPro-V [55]** | 57.2 | 36.0 | 51.6 | 43.2 | 45.6 | 23.2 | 42.8 | 61.2 | 35.2 | 53.2 | 47.6 | 48.4 | 19.2 | 44.1 |
| **Qwen-VL-Max [5]** | 62.4 | 28.8 | 49.2 | 45.2 | 37.2 | 21.2 | 40.7 | 62.4 | 36.0 | 54.8 | 46.4 | 38.0 | 24.0 | 43.6 |
| *Open-Source VLMs* | | | | | | | | | | | | | | |
| **InternVL-Chat-v1.5 [12]** | 59.2 | 33.2 | 51.2 | 42.8 | 40.4 | 30.4 | 42.9 | 65.6 | 42.4 | 56.4 | 50.8 | 46.8 | 28.0 | 48.3 |
| **InternLM-XComposer2 [18]** | 56.0 | 32.0 | **54.8** | 33.6 | 37.2 | 20.8 | 39.1 | 60.8 | 39.2 | 58.4 | 47.6 | 42.4 | 19.6 | 44.7 |
| **LLaVA-NeXT (Yi-34B) [35]** | **64.4** | 37.2 | 50.0 | 34.0 | 39.2 | 24.8 | 41.6 | 60.0 | 41.6 | 52.4 | 44.0 | 40.8 | 25.2 | 44.0 |
| **DeepSeek-VL-7B [39]** | 57.2 | 31.6 | 49.6 | 42.0 | 43.6 | 24.0 | 41.3 | 61.2 | 34.8 | 54.4 | 42.8 | 45.2 | 25.2 | 43.9 |
| **LLaVA-NeXT (Mistral-7B) [35]** | 60.8 | 30.8 | 49.6 | 37.6 | 36.8 | 20.4 | 39.3 | 62.8 | 36.4 | 53.6 | 42.0 | 35.2 | 26.4 | 42.7 |
| **MiniCPM-V-2 [47]** | 55.6 | 28.4 | 49.2 | 39.6 | 38.4 | 19.2 | 38.4 | 61.2 | 30.4 | 52.0 | 44.4 | 40.8 | 22.8 | 41.9 |
| **LLaVA-NeXT (Vicuna-13B) [35]** | 60.8 | 38.0 | 48.8 | 35.2 | 43.6 | 20.8 | 41.2 | 63.2 | 38.8 | 49.2 | 38.8 | 37.6 | 23.2 | 41.8 |
| **LLaVA-NeXT (Vicuna-7B) [35]** | 63.2 | 28.8 | 46.4 | 40.8 | 34.0 | 22.8 | 39.3 | 56.8 | 37.6 | 47.2 | 42.4 | 36.4 | 21.6 | 40.3 |
| **mPLUG-Owl2 [65]** | 47.2 | 24.0 | 45.2 | 33.6 | 28.8 | 18.8 | 32.9 | 53.2 | 34.4 | 45.2 | 38.8 | 38.8 | 23.6 | 39.0 |
| **LLaVA-v1.5-13B [34]** | 45.2 | 28.4 | 45.2 | 35.6 | 30.0 | 17.2 | 33.6 | 54.8 | 28.8 | 48.8 | 37.6 | 32.0 | 20.0 | 37.0 |
| **LLaVA-v1.5-7B [34]** | 48.8 | 29.6 | 48.0 | 34.4 | 27.6 | 17.2 | 34.3 | 50.8 | 31.6 | 50.4 | 38.8 | 32.0 | 25.2 | 38.1 |

Table 1: **Detailed Perception Performance on MMStar under Prism Evaluation Framework**. Reasoning module: ChatGPT. Abbreviations: Coarse Perception (CP), Fine-grained Perception (FP); Instance Reasoning (IR); Logical Reasoning (LR); Science&Technology (ST). Overall best scores are marked as **bold**, and intra-category best scores are marked as underline.

Unless otherwise specified, GPT-3.5-Turbo-0125 is adopted as the reasoning module. All evaluations are conducted using VLMEvalKit [14]. Further details on the evaluation are provided in Appx. B.1.

**Post-Processing.** Within the Prism framework, the LLM (particularly proprietary APIs) in the reasoning module often declines to answer questions due to insufficient clues in the extracted visual information. When Prism is employed as an evaluation framework, we classify this as a failure of perception for the visual question and refrain from any post-processing. For fair comparisons, whether Prism is contrasted with other VLMs as a vision-language task solver or different LLMs are evaluated as reasoning modules with varying rejection rates, a random choice is utilized as a fallback when option matching fails.

## 3.2 Main Results

We present the primary evaluation results of various VLMs' perception performance in Tab. 1. Results are categorized under two instruction types: *Generic Instruction* and *Query-Specific Instruction* .

**Generic Instruction Results.** When using generic instructions, GPT-4o exhibits exceptional performance across a variety of tasks. InternVL-Chat-v1.5 achieves the highest overall perception performance among open-source VLMs, nearly on par with the proprietary GPT-4v. LLaVA-NeXT (Yi-34B) demonstrates strong coarse and fine-grained perceptual abilities but lags behind GPT-4v in recognizing abstract elements in LR and ST visual questions. Smaller open-source VLMs, such as mPLUG-Owl2 and LLaVA-v1.5-7B, struggle with both coarse and fine-grained perception and encounter difficulties in recognizing essential elements in reasoning-related VQA.

**Query-Specific Instruction Results.** GPT-4o demonstrates significantly superior performance compared to all other models in the extraction and expression of visual information across all dimensions. GPT-4v is equally adept at handling coarse perception contents. Among open-source VLMs, InternVL-Chat-v1.5 excels in perception tasks across all dimensions excluding LR. Its performance not only surpasses that of other open-source VLMs but is also slightly ahead of GPT-4v. Occasionally, a decline in performance can be observed for specific VLM-task pairs when using query-specific instructions instead of generic ones, which can stem from difficulty in understanding query-specific instructions for specific domains.

## 3.3 Detailed Analysis

**Are Proprietary Models Getting Ahead of the Game?** When employing both generic and query-specific instructions, proprietary VLMs, particularly GPT-4o, significantly surpass other models in perceptual capabilities and can adeptly manage a wide range of tasks, as demonstrated in Tab. 1. Certain open-source models, such as InternVL-Chat-v1.5 and LLaVA-NeXT (Yi-34B), have achieved

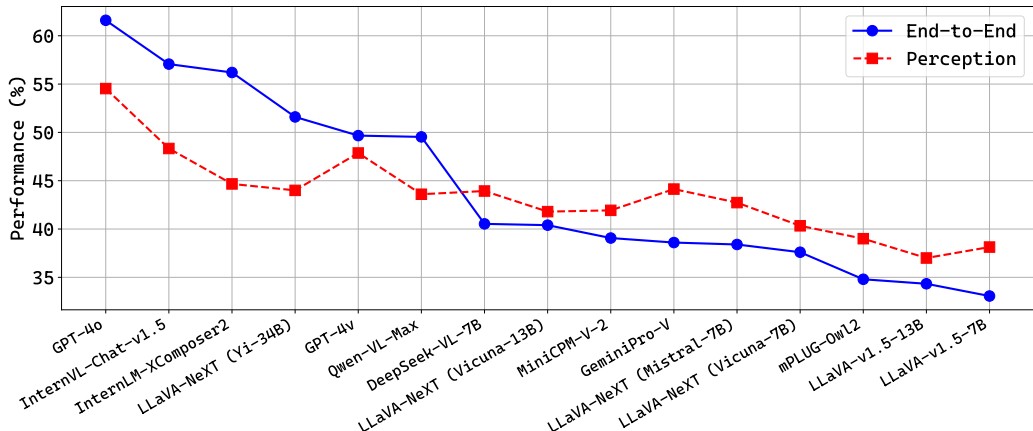

Figure 2: **Comparing End-to-End Performance and Perception Capability on MMStar.** We display model accuracies in end-to-end VQA and the Prism perception test with query-specific instructions. Most small-scale (7B, 13B, *etc.*) VLMs achieve better performance within Prism.

notable performance, approaching the capabilities of proprietary VLMs like GPT-4v and GeminiPro-V. Other open-source models, due to their limited perceptual abilities, generally perform slightly worse in Math and ST assessments. Notably, MiniCPM-V-2, a lightweight VLM with ~3B parameters, displays better perceptual performance compared to some 7B VLMs.

**The Gap between Perception Ability and End-to-End Performance.** In addition to solving visual questions in an end-to-end manner, Prism provides an alternative pipeline where the VLM is solely utilized for perception. The distinction between these two methods lies in the reasoning process: the former conducts reasoning internally within VLMs, whereas the latter performs reasoning based on VLM extracted information using an external LLM (ChatGPT). The comparison between these two approaches on MMStar is depicted in Fig. 2. For state-of-the-art large-scale VLMs such as GPT-4o and InternVL-Chat-v1.5, which are expected to possess excellent reasoning capabilities, employing an external ChatGPT for reasoning may diminish overall performance. Conversely, for most small-scale VLMs, using ChatGPT for reasoning significantly improves their performance, particularly in reasoning-related VQA, as shown in Fig. 5. This phenomenon indicates that the overall performance of small-scale VLMs can be heavily constrained by the size of the language model. To investigate whether the reasoning ability of ChatGPT constrains state-of-the-art VLMs, we implemented a Prism pipeline that decouples GPT-4o by using it as both the perception and reasoning module. The result reveals that this Prism pipeline, with post-processing, achieves an overall accuracy of 61%, nearly identical to the end-to-end GPT-4o performance of 61.6%.

**How does Language Model Size Affect Perception Ability?** During evaluation, we observe that the LLaVA-v1.5 series shows no significant improvement when using a larger language model (Vicuna-13B instead of Vicuna-7B, *etc.*). This suggests that perception performance may be independent of the language model size when using a relatively low-resolution vision backbone. However, the situation appears to differ with LLaVA-NeXT. Quantitative results for the LLaVA-NeXT series tell that scaling up the language model slightly enhances model perception, particularly when using query-specify instructions. Through a detailed qualitative analysis, we identified the primary factors contributing to the superior performance of larger LLaVA-NeXT models over smaller ones as follows: (1) **More Elaborate Expression**: Models equipped with a larger language encoder exhibit enhanced ability to articulate visual information. More detailed and organized narratives make it easier for the reasoning module to answer the question; (2) **More Adaptive to Instruction**: Larger language backbones entitle the model with a better understanding of instructions, yielding more suitable textual visual information for reasoning, particularly in response to query-specific instructions. In Fig. 3, we provide some qualitative results about the two typical modes.

### 3.4 Error Analysis

**Error Mode.** We conduct a thorough analysis and categorize errors in perception and reasoning into the following major modes:

1. **Major Error Modes in Perception**: (1) **Factual Errors.** VLMs may describe images with inaccuracies, such as stating that prominent elements are "not visible"; (2) **Incomplete Details.**

More Elaborate Expression

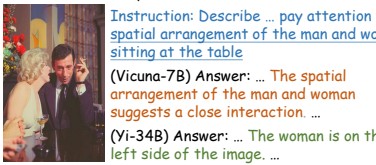

Instruction: Describe … pay attention to the spatial arrangement of the man and woman sitting at the table

(Vicuna-7B) Answer: … The spatial arrangement of the man and woman suggests a close interaction. …

(Yi-34B) Answer: … The woman is on the left side of the image. …

More Adaptive to Instruction

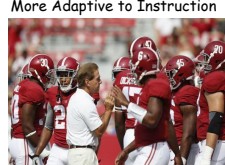

Instruction: Describe … pay attention to the length, texture, and color of the man's hair

(Vicuna-7B) Answer: … The coach, who is wearing a white shirt and khaki pants, is in the center of the group, clapping his hands. …

(Yi-34B) Answer: … The man's hair is short and appears to be a light brown color. The texture of his hair is straight, and it is neatly combed.. …

Figure 3: **The Effect of Language Model Size on Perception Ability.** We compare visual information extracted from different LLaVA-NeXT models. **Left**: LLaVA-NeXT (Yi-34B) tells the spatial arrangement in a more detailed way; **Right**: LLaVA-NeXT (Vicuna-7B) dismisses the query on the man's hair while LLaVA-NeXT (Yi-34B) tells all contents elaborately following the instruction.

| VLM | Agreement Rate (VLM) | Error Rate (VLM) |
|---|---|---|
| GPT-4o | 68.6 | 29.9 |
| GPT-4v | 61.5 | 38.0 |
| LLaVA-NeXT (Yi-34B) | 60.7 | 43.4 |
| LLaVA-v1.5-7B | 50.6 | 56.5 |

| LLM | Alignment Rate (LLM) |
|---|---|
| GPT-3.5-Turbo-0125 | 87.1 |
| DeepSeek-v2-Chat | 91.3 |
| Llama-3-70B-Instruct | 92.8 |
| GPT-4-Turbo-0125 | 90.7 |

Table 2: **Agreement Rates and Error Rates of Different VLMs.**

Table 3: **Alignment Rates of Different LLMs.**

Even in the absence of factual errors, VLMs may lack detailed content, resulting in insufficient information for reasoning; (3) **Instruction Following**: VLMs sometimes fail to follow instructions when providing corresponding descriptions.

2. **Major Error Modes in Reasoning**: (1) **Logical Errors.** LLMs may produce incorrect conclusions or reasoning processes due to limited reasoning abilities; (2) **Lack of Knowledge.** The absence of relevant domain knowledge prevents LLM from solving corresponding problems, especially in specific fields; (3) **Misunderstanding of the Query.** In rare cases, the query-specific part generated by LLMs deviates from the original question, misleading the perception of VLMs.

**Hallucination.** To reveal the source of hallucinations, we employ various LLMs to reason about descriptions generated by the given VLM and analyze the patterns in their results. We conduct experiments on various VLMs with a set of powerful language models, including GPT-3.5-Turbo-0125, GPT-4-Turbo-0125, Llama-3-70B-Instruct, and DeepSeek-v2-Chat. For each question, we gather results from four LLMs and consider cases where the same choice appears three times or more as "**agreement**". In agreement cases, we define the corresponding choice as "**voted choice**" and focus on the following cases:

**Case 1**: With descriptive texts generated by a specific VLM, the voted choice is wrong.

**Case 2**: With descriptive texts generated by a specific VLM, the voted choice is correct.

**Case 3**: For a specific LLM, it makes the same choice as the voted choice by all LLMs, while the voted choice is also the correct one.

For each VLM, we analyze the rate of case 1, which indicates the potential hallucinations of VLMs. For each LLM, we correspondingly calculate the rate of case 3 to case 2 to observe the alignment of its predictions with voted choices. The higher alignment rate means more robust reasoning since voted options are considered better. The notions are as follows.

$$\text{Agreement Rate (VLM)} = \frac{\text{number of agreement cases}}{\text{number of all cases}}$$

$$\text{Error Rate (VLM)} = \frac{\text{number of case 1}}{\text{number of agreement cases}}$$

$$\text{Alignment Rate (LLM)} = \frac{\text{number of case 3}}{\text{number of case 2}}$$

As presented in Tabs. 2 and 3, stronger VLMs show lower error rates, indicating that more capable VLMs experience fewer hallucination issues. By delving deep into the specific cases, we find that VLMs are prone to hallucinations in spatial awareness and fine-grained perception. All LLMs show relatively good alignment rates, demonstrating relatively robust reasoning performance. GPT-3.5-Turbo-0125 is more unstable by comparison. The cases where LLM's prediction does not align with the voted choice may stem both from reasoning ability issues and hallucinations.

| Instructions | GPT-4o | LLaVA-NeXT (Yi-34B) |
|---|---|---|
| Human 2 | 47.7 | 40.4 |
| GPT Synthesize 1 | 47.1 | 41.3 |
| GPT Synthesize 2 | 48.1 | 40.1 |
| CoT | 47.7 | 41.4 |
| Decompose | 47.3 | 41.4 |
| Human 1 | **48.4** | **41.6** |

Table 4: **Ablation on Different Generic Instructions.**

| Model | GPT-4o | GPT-4v | LLaVA-NeXT (Yi-34B) | LLaVA-v1.5-7B |
|---|---|---|---|---|
| GPT-3.5-Turbo-0125 | 54.7 | 48.5 | 44.1 | **38.5** |
| GPT-4-Turbo-0125 | 56.9 | 48.7 | **48.4** | 38.3 |
| Llama-3-70B-Instruct | **59.3** | **51.3** | 47.7 | **38.5** |
| DeepSeek-v2-Chat | 57.8 | 49.7 | 45.8 | 35 |

Table 5: **Ablation on Using Different LLMs as the Reasoning Module.**

---

**Different Generic Instructions**

**Human 1:** Describe the fine-grained content of the image, including scenes, objects, relationships, instance location, and any text present.

**Human 2:** Describe the fine-grained content of the image, including scenes, objects, relationships, instance location, background and any text present. Please skip generating statements for non-existent contents and describe all you see.

**GPT Synthesize 1:** Given the image below, please provide a detailed description of what you see.

**GPT Synthesize 2:** Analyze the image below and describe the main elements and their relationship.

**CoT:** Describe the fine-grained content of the image, including scenes, objects, relationships, instance location, and any text present. Let's think step by step.

**Decompose:** Decompose the image into several parts and describe the fine-grained content of the image part by part, including scenes, objects, relationships, instance location, and any text present.

Figure 4: **Different Generic Instructions we adopted in the Ablation Study.**

# 4 Ablation Study

**Ablation on Generic Instructions.** Within Prism, the generic instructions for visual information extraction are crucial. We experimented with a variety of instructions to elicit the fundamental perceptual capabilities of VLMs, including human-written instructions, GPT-4 generated instructions, and those incorporating chain-of-thought [61] or explicit decomposition, as shown in Fig. 4. We conducted an ablation study on MMStar using the state-of-the-art VLM GPT-4o and LLaVA-NeXT (Yi-34B). As illustrated in Tab. 4, `Human 1` outperforms others in eliciting the fundamental perceptual capabilities of the models, while the differences among various instructions are not significant. Therefore, we adopt `Human 1` as the generic instruction for all evaluations.

**Ablation on the Reasoning Module.** The reasoning module is critical for accurately determining the correct answer based on the visual information. To evaluate the impact of the reasoning module on overall performance, we select four LLMs: `GPT-3.5-Turbo-0125`, `GPT-4-Turbo-0125`, `Llama-3-70B-Instruct`, and `DeepSeek-v2-Chat`, and assess these models across four VLMs with varying capacities. The comparative results are presented in Tab. 5. As a freely available model, GPT-3.5 demonstrates good reasoning performance under our framework, and the more advanced GPT-4 shows improved performance in line with other benchmarks. Notably, `Llama3-70B-Instruct`, as a representative of open-source models, exhibits competitive capabilities compared to `GPT-4-Turbo-0125` under different perceptual conditions. This suggests that open-source models could be valuable for further exploration in visual reasoning.

**Ablation on the Vision Backbone.** To investigate the impact of the vision encoder on perception ability within the LLaVA architecture, we conduct an ablation study on three pre-trained visual backbones, including `CLIP ViT-L/14`, `SigLip-SO400M`, and `InternViT-6B`. We use `InternLM2-7B` as the fixed language encoder in LLaVA and train the VLMs for one epoch with the vision encoder fixed. The results in Tab. 6 show that `SigLip-SO400M` achieves better performance compared to `CLIP ViT-L/14` and `InternViT-6B` on MMStar.

# 5 PrismCaptioner

Within Prism, we explore the use of small-scale Vision-Language Models (VLMs) as a perception module. We use `SigLip` as the vision encoder, `InternLM2-[1.8B/7B]` as the language encoder to develop two visual captioners at different scales, refered as `PrismCaptioner-[2B/7B]`.

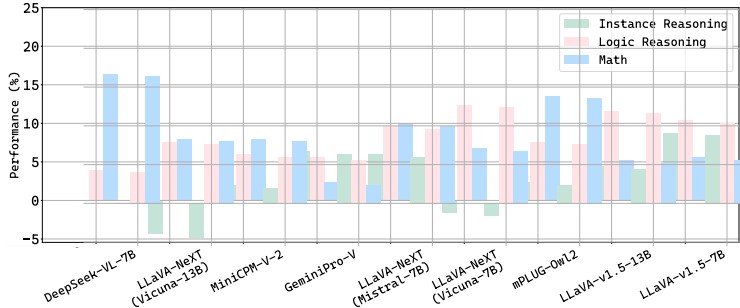

| Vision Backbone | Overall |
|---|---|
| **SigLip-SO400M** | **43.5** |
| **InternViT-6B** | 42.7 |
| **CLIP ViT-L/14** | 41.7 |

Figure 5: **The Performance Changes of Using an External LLM (ChatGPT) for Reasoning of Small Scale VLMs.**

Table 6: **Ablation on the Vision Backbone.**

## 5.1 Training Details

We perform one-stage training for two epochs using ZeRO2 with XTuner [15] on 8 NVIDIA A800-80GB GPUs, and the training lasts less than a day. The training data include of one copy of `ALLaVA-Caption-4V` and two copies of `Evol-Intruct-GPT4-Turbo-143K`. The batch size is set to 16 per device for `PrismCaptioner-2B` and 8 for `PrismCaptioner-7B`. We utilized the AdamW optimizer and 2e-4 learning rate, with the warm-up ratio set to 0.03 and $(\beta_1, \beta_2)$ set to (0.9, 0.999). No weight decay was applied and a maximum norm value of 1 is applied for gradient clipping. Full details about QLoRA are presented in Appx. B.2.

## 5.2 The Performance of PrismCaptioner

We conduct a thorough evaluation of `PrismCaptioners` across multiple benchmarks, employing `GPT-3.5-Turbo-0125` and `Llama-3-70B-Instruct` as the reasoning module. We utilize MMStar as our primary benchmark to assess comprehensive multimodal capabilities. For domain-specific evaluations, we choose AI2D to gauge diagram comprehension, MMMU for expert knowledge assessment, and MathVista to test mathematical proficiency. The results are presented in Tab. 7. In line with the benchmark selection principles outlined in Sec. 2.2, we apply a consistent filtering strategy across AI2D, MMMU, and MathVista, mirroring that of MMStar, to retain only vision-essential and uncontaminated questions, denoted by the suffix (F). In addition to comparisons with existing open-source and proprietary VLMs, we also assess two baseline models, `LLaVA-InternLM2-[1.8B/7B]`, which are `PrismCaptioners` trained on LLaVA-v1.5 instruction tuning data. We also compare `PrismCaptioners` and `ShareCaptioner`, an open-source VLM designed for generating informative image captions, under identical Prism framework configurations.

As depicted in Tab. 7, each `PrismCaptioner`, with an external potent LLM for reasoning, markedly surpasses its corresponding end-to-end baseline. `PrismCaptioners` also outperform `ShareCaptioner` across all multimodal benchmarks. For the 7B variant, the integration of `Llama3` results in a substantial enhancement, positioning `PrismCaptioner-7B` as a highly competitive vision-language solver, particularly on MMStar and MMMU. For `PrismCaptioner-2B`, employing `ChatGPT` yields superior results, outperforming nearly all 7B VLMs in general aptitude, expert knowledge, and mathematical skills. Remarkably, it achieves performance levels on par with some ten times larger VLMs, such as `LLaVA-InternLM2-20B`, `Yi-VL-34B`, and `Emu2-Chat`. This demonstrates that Prism enables the creation of a robust yet efficient vision-language solver, exemplified by `PrismCaptioner-2B` with `ChatGPT`, which delivers impressive results.

## 6 Discussion

**Prism's Value as an Evaluation Framework.** Prism's value as an Evaluation Framework lies in its ability to disentangle and measure the perception and reasoning capabilities of VLMs across various data sources. There exist specialized multimodal benchmarks designed to assess VLMs' perception and reasoning capabilities, yet they often focus on specific domains. For instance, RealWorldQA [62] evaluates real-world perception with high-resolution images, OCRVQA [44] assesses text recognition in publications, and POPE [30] determines object existence in images. However, many interested domains (geometry, medical images, GUIs, *etc.*) are not covered by those perception benchmarks. Prism fills this gap by enabling the measurement and comparison of VLMs' perception capabilities on

| Model | MMStar | MMMU | MMMU (F) | MathVista | MathVista (F) | AI2D | AI2D (F) |
|---|---|---|---|---|---|---|---|
| *Proprietary VLMs* | | | | | | | |
| **GPT-4o [46]** | **61.6** | **62.8** | **45.9** | 56.5 | 46.5 | **82.2** | **59.7** |
| **GPT-4v [46]** | 49.7 | 53.8 | 42.0 | 48.7 | 32.0 | 75.9 | 45.7 |
| *Open-Source VLMs* | | | | | | | |
| **InternVL-Chat-v1.5 [12]** | 57.1 | 46.8 | 33.7 | 54.7 | 47.5 | 80.6 | 55.0 |
| **InternLM-XComposer2 [18]** | 56.2 | 41.4 | 21.2 | **59.5** | **49.0** | 81.2 | 57.5 |
| **LLaVA-NeXT (Yi-34B) [35]** | 51.6 | 48.8 | 26.7 | 40.4 | 29.8 | 78.9 | 51.6 |
| **LLaVA-NeXT (Vicuna-13B) [35]** | 40.4 | 37.3 | 16.5 | 34.1 | 19.6 | 72.2 | 36.2 |
| **LLaVA-NeXT (Mistral-7B) [35]** | 38.4 | 37.0 | 19.2 | 34.6 | 21.3 | 69.0 | 32.3 |
| **LLaVA-NeXT (Vicuna-7B) [35]** | 37.6 | 37.6 | 18.0 | 31.5 | 17.1 | 67.0 | 29.3 |
| **Yi-VL-34B [1]** | 40.5 | 45.1 | 21.2 | 31.5 | 12.0 | 65.9 | 26.4 |
| **Emu2-Chat [53]** | 40.7 | 35.0 | 27.8 | 30.7 | 14.0 | 49.7 | 22.7 |
| **LLaVA-InternLM2-20B [15]** | 41.9 | 39.4 | 18.0 | 25.3 | 9.9 | 65.4 | 28.4 |
| **DeepSeek-VL-7B [39]** | 40.5 | 38.3 | 19.6 | 36.9 | 20.7 | 65.3 | 36.9 |
| **MiniCPM-V-2 [47]** | 39.1 | 38.2 | 21.2 | 39.8 | 25.4 | 62.9 | 26.9 |
| **LLaVA-v1.5-13B [34]** | 34.3 | 37.0 | 15.3 | 27.7 | 10.3 | 61.1 | 23.0 |
| **LLaVA-v1.5-7B [34]** | 33.1 | 35.7 | 15.7 | 25.6 | 8.5 | 55.5 | 17.8 |
| **mPLUG-Owl2 [65]** | 34.8 | 34.7 | 19.6 | 25.4 | 8.5 | 55.7 | 20.5 |
| *Open-Source VLMs (E2E Baselines)* | | | | | | | |
| **LLaVA-InternLM2-1.8B [15]** | 34.5 | 30.2 | 18.4 | 26.3 | 9.1 | 43.6 | 23.5 |
| **LLaVA-InternLM2-7B [15]** | 38.3 | 40.1 | 21.9 | 26.0 | 7.4 | 63.6 | 26.6 |
| *Prism Models* | | | | | | | |
| **ShareCaptioner-**`ChatGPT` | 38.7 | 45.2 | 30.2 | 30.2 | 12.6 | 59.6 | 22.2 |
| **PrismCaptioner-2B-**`ChatGPT` | 43.3 | 46.6 | 32.0 | 33.5 | 18.2 | 62.0 | 27.4 |
| **PrismCaptioner-2B-**`Llama3` | 42.0 | 46.7 | 32.4 | 33.5 | 19.8 | 59.8 | 30.3 |
| **PrismCaptioner-7B-**`ChatGPT` | 43.7 | 47.3 | 29.8 | 35.1 | 24.6 | 65.4 | 30.6 |
| **PrismCaptioner-7B-**`Llama3` | 45.9 | 53.3 | 35.6 | 39.0 | 27.3 | 68.1 | 37.2 |

Table 7: **Detail Results of Models under Prism Framework.**[1] (F) represents the sub-dataset filtered by our strategy in order to ensure vision indispensability and avoid data leakage. The suffix of Prism models (`ChatGPT` or `Llama3`) indicates the reasoning module adopted.

general VQA datasets in these domains. Additionally, existing 'reasoning' benchmarks [40, 23] are compositional, requiring VLM to recognize key elements and before reasoning. Comparing an end-to-end VLM with Prism equipped with the same VLM and an external LLM, like ChatGPT, provides insights into the VLM's intrinsic reasoning capabilities and potential performance constraints.

**Prism's Value as a Vision-Language Solver.** By integrating small-scale VLMs with readily available external LLMs within Prism, we achieve superior performance compared to the standard end-to-end capabilities of the standalone VLM. This approach also renders it practical to address vision-language tasks using a 2B

| VLM | CP | FP | IR | LR | Math | ST | Overall |
|---|---|---|---|---|---|---|---|
| **GPT-4v** | 67.2 | 43.6 | 56.8 | 46.8 | 42.0 | 30.8 | 47.9 |
| **GeminiPro-V** | 61.2 | 35.2 | 53.2 | 47.6 | 48.4 | 19.2 | 44.1 |
| **Ensemble** | 66.4 | 47.2↑ | 58.8↑ | 50.8↑ | 46.4 | 32.0↑ | 50.3↑ |

Table 8: **Performance of Perception Module with Multiple VLMs on MMStar.**

parameter VLM, as reasoning is effectively outsourced to external LLMs. When implemented with an LLM API, Prism's inference process (without quantization) only consumes several gigabytes of GPU memory. Furthermore, Prism allows for the flexible incorporation of multiple VLMs to enhance perception. For instance, the straightforward concatenation of outputs from GPT-4v and GeminiPro-V has demonstrated substantial improvements across the majority of metrics on the MMStar benchmark, as substantiated by the data presented in Table 8.

---

[1]For comparison, we use single image as input and limit the maximum numbers of output tokens to 512. Better performance of `PrismCaptioners` is presented in Appx. B.3

## Acknowledgement

This project is supported by the National Key R&D Program of China (No.2022ZD0161600), the Shanghai Postdoctoral Excellence Program (No.2023023), China Postdoctoral Science Fund (No.2024M751559), and Shanghai Artificial intelligence Laboratory. This project is funded in part by the Centre for Perceptual and Interactive Intelligence (CPII) Ltd under the Innovation and Technology Commission (ITC)'s InnoHK. Dahua Lin is a PI of CPII under the InnoHK.

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

# A Related Work

## A.1 Large Vision-Language Models (LVLMs)

The landscape of Large Language Models (LLMs) is continually evolving, with an expanding body of research focused on integrating multimodal capabilities to enhance their perceptual abilities in real-world contexts [49, 70, 29]. Early efforts in this direction, such as Flamingo [3], introduced gated dense blocks of cross-attention within pre-trained language encoder layers to fuse visual features. Subsequent models like BLIP2 [29] and InstructBLIP [16] utilized a Q-former to align features across different modalities, enabling tasks such as zero-shot visual question answering. More recent models, including LLaVA [36] and MiniGPT-4 [70], have simplified modality bridging through the use of MLP-based projection layers, d offering a more straightforward approach compared to the Q-former. The architecture of LLaVA has been widely adopted in subsequent works [35, 1, 12, 39]. The choice of vision encoders and the language model is considered critical for the overall performance of VLMS. Most VLMs [36, 18, 35, 27, 70] employ CLIP-based Vision Transformer [50, 67, 54] as the vision encoder, owing to its good pre-training alignment of visual and textual modalities. There is a prevalent belief [37, 35] that the scale of the language model significantly impacts the performance of VLMs, though detailed analyses are lacking. In addition to the open-source VLMs developed by the academic community, numerous proprietary VLMs [46, 55, 6, 48] demonstrate robust performance across various multimodal benchmarks. This paper presents a breakdown capability analysis of both open-source and proprietary VLMs using the Prism framework.

## A.2 LVLMs Capability Evaluation

Large-scale VLMs have demonstrated promising outcomes across a diverse range of multimodal tasks, as evidenced by extensive qualitative and quantitative evaluations. Early assessments of LVLMs often involved open-ended Visual Question Answering (VQA) [2, 24, 41, 22] and human-based subjective evaluations [64, 63]. However, these methods face limitations in accurately reflecting the true performance of VLMs. Open-ended VQA tasks typically demand an exact match between the model's prediction and the ground truth, which can lead to a significant number of false positives. Conversely, subjective evaluations introduce biases and make the results challenging to reproduce. Subsequent research has shifted towards structuring visual questions in closed-ended formats, such as multiple-choice or Yes-or-No questions. Pioneering works like MMBench [37], MME [20], or SEED-Bench [28] have presented comprehensive evaluations of VLMs using closed-ended VQA, covering various perception and reasoning capabilities. Additionally, specialized multimodal benchmarks have emerged to assess VLMs from specific angles. For instance, MMMU [66] evaluates VLMs' ability to handle multimodal examination questions, while POPE [30] and HallusionBench [33] scrutinize hallucination and illusion phenomena in VLMs. RealWorldQA [62] focuses on real-world perception with high-resolution images. Recently, MMStar [10] has addressed the issues of vision dispensability and data contamination in existing benchmarks by compiling high-quality, vision-indispensable questions from multiple sources, ensuring minimal data leakage and covering six core capabilities. In this study, Prism primarily utilizes MMStar for capability evaluation.

## A.3 LVLMs Capability Breakdown Investigation

To offer insightful and detailed feedback for the future optimization of VLMs, some researchers have initiated efforts to dissect the abilities of VLMs, aiming to uncover strategies for enhancement. To more effectively explore the disparity between VLMs and human cognition, Zhang et al. [69] employ Raven's Progressive Matrices (RPM) to examine the model's deductive reasoning skills grounded in visual perception. Through error case analysis, the authors observe that VLMs often make compounding and confounding errors when articulating individual elements within RPM, which subsequently results in erroneous reasoning. Although this study provides qualitative insights, it does not encompass a systematic evaluation of perception and reasoning capabilities. Meanwhile, researchers have developed specialized benchmarks to assess specific capabilities. For example, InfiMM-Eval [23] and MathVista [40] have implemented rigorous, step-by-step evaluations of the model's complex reasoning abilities on natural images and mathematical VQA problems, respectively. However, these reasoning benchmarks require a foundational perception capability to accurately identify key elements. Concurrently, there are perception benchmarks [58, 62, 30, 21] that exclusively focus on evaluating the perception skills of VLMs across various scenarios. Moreover, some

specific crucial capabilities of VLMs have received researchers' attention, such as fine-grained understanding [68] and visio-linguistic compositional knowledge [57, 68, 25]. In this study, Prism introduces a general decoupling framework that facilitates a detailed analysis of perception and reasoning capabilities, applicable to any multimodal benchmark.

## B   Supplementary Details of Prism

### B.1   Details of Prism Evaluation Framework

#### B.1.1   Query-Specific Instruction Details

*Query-Specific Instruction* is a combination of generic instruction and query-specific part, as depicted in Fig. 6. To ensure that the query-specific part generated by the reasoning module is closely related to the questions and options, we adopt few-shot learning to guide the LLM. For each visual question, we feed the request with multiple examples. This approach helps the reasoning module understand what "contents to observe" means in different contexts and allows it to make accurate inferences in response to specific questions, as illustrated by the prompts in Fig. 7.

---

**Instructions**

**Generic Instruction:** Describe the fine-grained content of the image, including scenes, objects, relationships, instance location, and any text present.
**Query-Specific Instruction:** Describe the fine-grained content of the image, including scenes, objects, relationships, instance location, and any text present. Especially, pay attention to <query-specific part>.

---

Figure 6: **Generic Instruction *vs*. Query-Specific Instruction.**

---

**The Few-shot Prompt Template**

Your task is to give a concise instruction about what basic elements are needed to be described based on the given question. Ensure that your instructions do not cover the raw question, options, or thought process of answering the question.

**Question:** In which period the number of full-time employees is the maximum?
**Contents to observe:** the number of full-time employees
**Question:** What is the value of the smallest bar?
**Contents to observe:** the heights of all bars and their values
**Question:** What is the main subject of the image?
**Contents to observe:** the central theme or object
**Question**: What is the position of the catcher relative to the home plate?
**Contents to observe:** the spatial arrangement of the objects
**Question:** What is the expected ratio of offspring with white spots to offspring with solid coloring? Choose the most likely ratio.
**Contents to observe:** the genetic information
**Question:** <question>
**Contents to observe:**

---

Figure 7: **The Prompt Template for the Reasoning Module to Generate the "Contents to Observe" Part.**

#### B.1.2   Inference Prompt Template of the Reasoning Module

After the perception module generates detailed visual information about the image, the query needs to be reformatted to enable the reasoning module to answer more accurately based on the information

| Hyperparameter | Assignment |
|---|---|
| Image Resolution | 384 * 384 |
| Patch Size | 14 |
| Max Length | 1296 |
| QLoRA Quantization | 4bit |
| LLM int8 Threshold | 6.0 |
| BnB 4bit Compute Dtype | torch.float16 |
| BnB 4bit Double Quant | True |
| BnB 4bit Quant Dtype | nf4 |
| LoRA r | 512 |
| LoRA $\alpha$ | 256 |
| LoRA Dropout | 0.05 |

Table 9: **PrismCaptioner Training Details.**

and the question. The template for reformatting is shown in Fig. 8. The reformatting process involves simple splicing, making it intuitive for the reasoning module to respond directly. An example of this can be seen in Fig. 9.

---

**Reformatting Template**

You are an excellent text-based reasoning expert. You are required to answer the question based on the detailed description of the image.
**Description:** <description>
**Question:** <question>

---

Figure 8: **The Template for Reformatting Query.**

---

**Reformatted Query Example**

You are an excellent text-based reasoning expert. You are required to answer the question based on the detailed description of the image.
**Description:** The image presents a delightful celebration of Father's Day. Dominating the center of the image is a blue tie, adorned with white stripes, symbolizing the essence of fatherhood. The tie is slightly tilted to the right, adding a touch of dynamism to the composition. On the left side of the tie, the phrase "Happy Father's Day" is elegantly inscribed in a white cursive font, extending warm wishes to all dads. The text and the tie are set against a dark blue background, creating a striking contrast that draws attention to the main elements of the image. Adding a final touch of sophistication, a thin white border frames the entire image, encapsulating the joyous message of Father's Day. The image, in its entirety, serves as a heartfelt tribute to all the wonderful fathers out there.
**Question:** Which special day is associated with this poster?
Options: A. Earth Day. B. National Reading Day. C. Father's Day. D. Mother's Day
Please select the correct answer from the options above.

---

Figure 9: **An Example of the Reformatted Query.**

## B.2 More Details of PrismCaptioner Training

The details about QLoRA for training PrismCaptioner is demonstrated in Tab. 9

| Model | MMStar | MMMU | MMMU (F) | MathVista | MathVista (F) | AI2D | AI2D (F) |
|---|---|---|---|---|---|---|---|
| *Prism Models (Multiple Image Inputs & Max Output Length 2048)* | | | | | | | |
| **PrismCaptioner-2B**-ChatGPT | 43.0 | 46.1 | 31.4 | 34.8 | 16.7 | 62.1 | 27.9 |
| **PrismCaptioner-2B**-Llama3 | 42.6 | 49.0 | 34.1 | 35.1 | 19.8 | 59.8 | 30.3 |
| **PrismCaptioner-7B**-ChatGPT | 43.4 | 47.9 | 33.7 | 36.5 | 25.0 | 65.4 | 30.8 |
| **PrismCaptioner-7B**-Llama3 | 46.2 | 56.5 | 42.0 | 39.8 | 26.2 | 67.9 | 37.7 |

Table 10: **More Detailed Performance Results of PrismCaptioners**

## B.3 More Performance Details of PrismCaptioner

In addition to the results of Tab. 7, we use multiple images as iputs if there are and set maximum output length to 2048. The performace is presented in Tab. 10

## C Detailed Examples of Prism

### C.1 End-to-end *v.s.* Prism Predictions

Some end-to-end models with small language models, often predict incorrect answers due to their limited reasoning capabilities. Utilizing an external LLM, Prism endows VLMs with the reasoning ability to solve the vision-language tasks, as presented in Tab. 11 and Tab. 12.

### C.2 PrismCaptioner Performances

PrismCaptioner can professionally extract and express detailed visual information to solve coarse perception, fine-grained perception, instance reasoning, logical reasoning, science & technology, and math tasks, as presented in Tabs. 13 to 18.

### C.3 Performances Comparison between PrismCaptioner and GPT-4o

Although PrismCaptioner can generate detailed descriptions of images, there is still room for improvement compared to GPT-4o.

(1) GPT-4o generates descriptions that are more relevant to the questions. Due to limited training data, PrismCaptioner's captions are less adaptive to query-specific parts compared to those of GPT-4o, as shown in Tab. 19.

(2) GPT-4o's expression is more detailed and specific, and the responses about spatiotemporal information are also more accurate, as shown in Tab. 20.

## D Limitations and Broader Impacts

**Limitations.** In this study, we introduce the Prism framework and showcase its effectiveness as both an analytical tool and a versatile vision-language task solver. Prism relies on language descriptions while natural languages may struggle to express some obscure visual elements, especially some aesthetic contents like abstract artworks and surrealist paintings. It is difficult to obtain high-quality descriptions in these cases. Given budget constraints, our evaluation focuses on a select group of representative open-source and proprietary VLMs, which may not encompass all the most recent high-performing models. Our experiments with training visual captioners aim to illustrate Prism's capability to deliver strong performance on vision-language tasks while minimizing costs. To fully realize the potential of Prism, additional experiments are recommended, particularly on domain-specific visual instruction tuning data, such as tables and diagrams, screens, and graphical user interfaces (GUIs), medical images, etc., to thoroughly assess Prism's efficacy in these specialized contexts.

**Broader Impacts.** As an analytical framework, Prism offers detailed insights into the perception and reasoning capabilities of vision-language models (VLMs), providing valuable guidance for future model optimization. Training large-scale VLMs necessitates extensive multi-modal data and

**Question: What is the primary scene depicted in the image?**
**Image:**

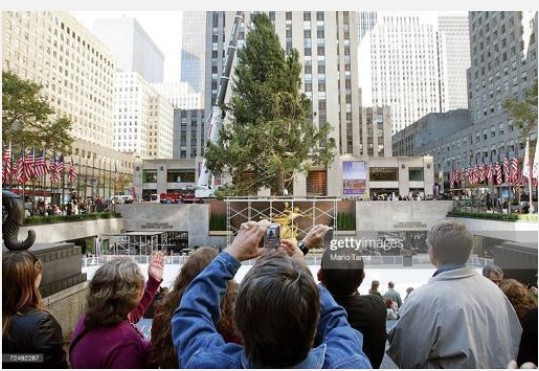

**Options:**
A. A group of people taking pictures of a statue in a plaza
B. A large crowd taking pictures of each other in an outdoor event
C. A group of people taking pictures of a Christmas tree in a plaza
D. A group of people taking pictures of a famous building in New York City
**Category:** Coarse Perception
**Model:** `LLaVA-NeXT (Vicuna-7B)`
**End-to-end Prediction:** D. A group of people taking pictures of a famous building in New York City ( ✖ )
**Perception Description:** The image captures a bustling scene in **a city square**, where **a large crowd of people** has gathered. The crowd is diverse, with individuals of various ages and attire, all seemingly engaged in the event taking place. The central focus of the image is a large, **illuminated Christmas tree**, which stands prominently in the square. The tree is adorned with a multitude of colorful lights and ornaments, creating a festive atmosphere.
In the background, the cityscape is visible with tall buildings, some of which are adorned with American flags, suggesting a location in the United States. The flags are flying at half-mast, adding a solemn tone to the scene.
The sky above is overcast, casting a soft light over the scene. The overall mood of the image is one of celebration and unity, as the crowd seems to be enjoying the holiday festivities. The image does not contain any text. The relative positions of the objects suggest a well-organized event, with the Christmas tree as the focal point and the crowd surrounding it. The flags in the background provide a sense of location and national pride. The overcast sky adds a touch of solemnity to the otherwise joyful scene.
**Perception Prediction:** C. ( ✔ )

Table 11: **An Example of End-to-End *v.s.* Prism `LLaVA-NeXT (Vicuna-7B)` Prediction.**

computational resources. Prism mitigates this challenge by training small-scale VLMs that specialize in visual captioning tasks and perform reasoning with large language models (LLMs), which are now cost-effective.[2] When employed as a vision-language task solver, Prism can be trained and deployed at a significantly reduced cost, making it a promising approach for highly customized applications or tasks with limited training data. However, Prism also carries potential societal impacts, as it could lower the barrier to building multimodal applications, some of which may be harmful. There is a risk that Prism could be used to develop harmful multimodal AI systems. Additionally, data-driven methods often inherit biases, which can persist in downstream tasks. We urge users to thoughtfully consider the implications of these biases when implementing our model.

---

[2]Thanks to the development in deploying technologies, the inference of LLM is now cheap, provided by various corporations at a price as low as millions of tokens per US dollar.

**Question: Which number is missing?**
**Image:**

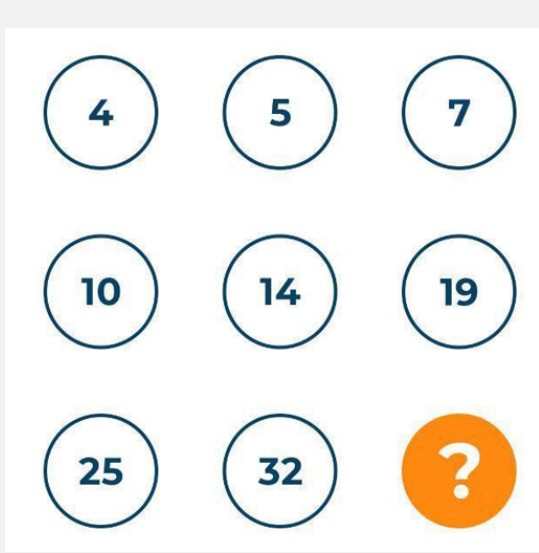

**Options:** A. 40    B. 10    C. 22    D. 34
**Category:** Logical Reasoning
**Model:** MiniCPM-V-2
**End-to-end Prediction:** B. 10 ( ✖ )
**Perception Description:** The image presents a grid of nine circles, each containing a number from 4 to 25. The numbers are arranged in a 3x3 grid, with the numbers 4, 5, 7, 10, 14, 19, 25, 32, and a question mark (?) at the bottom right corner. The sequence of numbers in the grid is 4, 5, 7, 10, 14, 19, 25, 32, with the question mark indicating the missing number.
**Perception Prediction:** The pattern in the sequence is as follows: 5 - 4 = 1, 7 - 5 = 2, 10 - 7 = 3, 14 - 10 = 4, 19 - 14 = 5, 25 - 19 = 6, 32 - 25 = 7. Therefore, the missing number should be 32 + 8 = 40.
So, the correct answer is: A. 40 ( ✔ )

Table 12: **An Example of End-to-End** *v.s.* **Prism** MiniCPM-V-2 **Prediction.**

**Question: Which image is more brightful?**
**Image:**

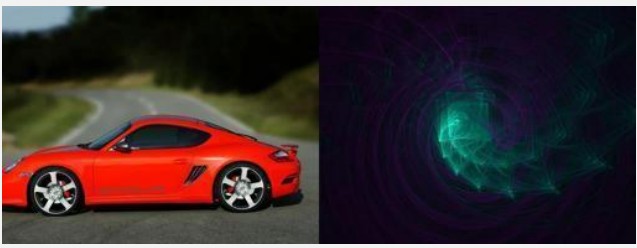

**Options:** A. The first image    B. The second image
**Category:** Coarse Perception
**Description:** The image is a split composition with two distinct halves.
On the left side, there is a photograph of a **bright** red sports car parked on the side of a road. The car appears to be a modern coupe with a sleek design, featuring a long hood and a short rear deck, which are characteristic of high-performance vehicles. It has five-spoke alloy wheels with a silver finish and low-profile tires. The car's windows are tinted, and it has a visible antenna on the rear part of the roof. The background shows a blurred natural landscape with greenery, suggesting a rural or semi-rural setting.
On the right side of the image, there is a digital artwork depicting a swirling, fractal-like pattern in shades of green and purple, set against a **dark** background. The pattern has a hypnotic, almost hypnotic quality, with a sense of motion and energy emanating from it. The artwork has a hypnotic, almost hypnotizing effect, drawing the viewer's attention to the center of the swirl where the colors seem to converge.
**Prediction:** A. The first image ( ✔ )

Table 13: **PrismCaptioner Generates Description to Solve Coarse Perception Task.**

**Question: What color is the shirt of the man playing the guitar?**
**Image:**

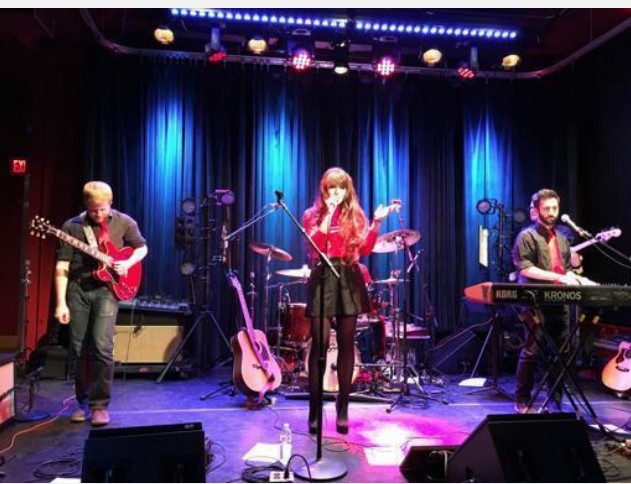

**Options:** A. Blue    B. Brown    C. Red    D. Black
**Category:** Fine-grained Perception
**Description:** The image depicts a live music performance by a band on stage. There are three members visible: a vocalist, a guitarist, and a keyboard player. The vocalist, positioned in the center, is a woman with long hair, wearing a red top and a black skirt. She is holding a microphone stand with her right hand and appears to be singing into it. To her left, a man is playing an electric guitar, dressed in a **black shirt** and jeans, focused on his instrument. To the right of the vocalist, another man is playing a keyboard, wearing a black shirt and dark pants. The stage is equipped with various musical instruments and equipment, including a drum set behind the vocalist, a keyboard in front of the keyboard player, and a microphone stand in front of the vocalist. The lighting is dim, with spotlights highlighting the performers, and the background is a draped curtain in shades of blue and red. The atmosphere suggests an intimate concert setting.
**Prediction:** D. Black ( ✔ )

Table 14: **PrismCaptioner Generates Description to Solve Fine-Grained Perception Task.**

**Question: Which solution has a higher concentration of pink particles?**
**Image:**

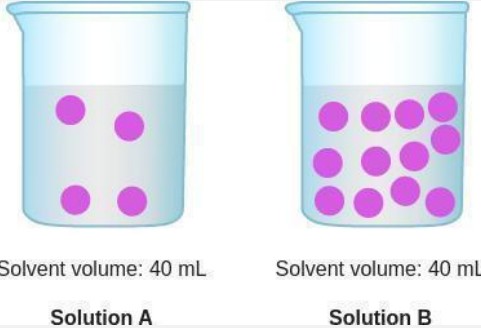

**Options:** A. neither; their concentrations are the same     B. Solution A     C. Solution B
**Category:** Instance Reasoning
**Description:** The image displays two beaker-like containers, labeled Solution A and Solution B, each containing a clear liquid. Solution A has a volume of 40 mL, while Solution B has a volume of 40 mL. Inside Solution A, there are multiple small, round, purple particles dispersed uniformly throughout the liquid. In contrast, Solution B contains a similar number of purple particles, but they are clustered **more densely**, suggesting a higher concentration. The background is plain white, focusing attention on the containers and their contents. There are no additional background elements or context provided.
**Prediction:** C. Solution B ( ✔ )

Table 15: **PrismCaptioner Generates Description to Solve Instance Reasoning Task.**

**Question: In nature, what's the relationship between these two creatures?**
**Image:**

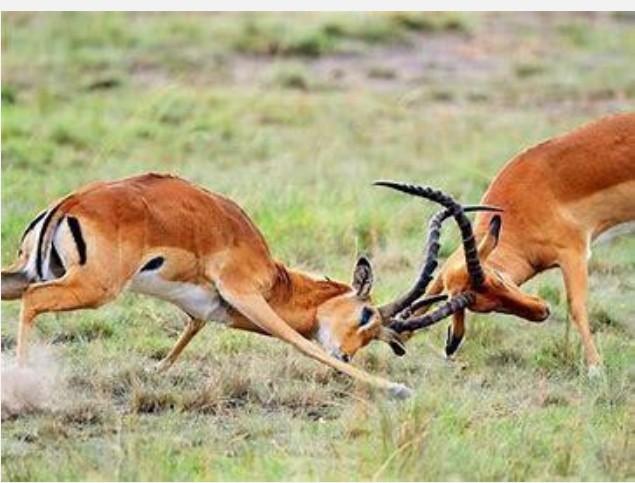

**Options:**
A. Predatory relationships     B. Competitive relationships
C. Parasitic relationships     D. Symbiotic relationship
**Category:** Logical Reasoning
**Description:** The image depicts two antelopes engaged in what appears to be **a physical confrontation** on a grassy savanna. The antelope on the left is rearing up on its hind legs, with its front legs extended towards the other antelope, which is also on its hind legs, facing the first antelope. Both animals have prominent, curved horns, and their fur is a rich, reddish-brown color. The background is a blurred natural landscape, suggesting a wild, open environment typical of a savanna ecosystem. There are no human elements or artificial structures visible, reinforcing the natural setting. The colors are vibrant, with the green of the grass contrasting with the reddish-brown of the antelopes.
**Prediction:** B. Competitive relationships ( ✔ )

Table 16: **PrismCaptioner Generates Description to Solve Logical Reasoning Task.**

**Question:** What comes after the full moon?
**Image:**

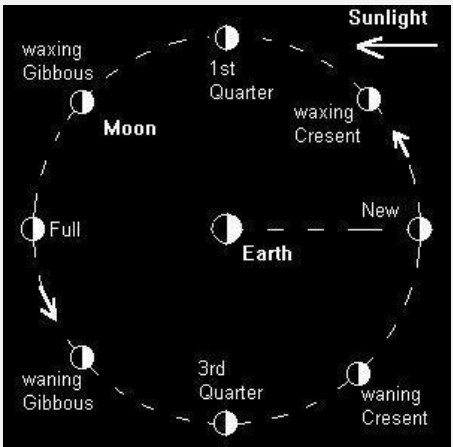

**Options:** A. waning cresent   B. 3rd quarter   C. waning Gibbous   D. new moon
**Category:** Science & Technology
**Description:** The image depicts a simplified diagram illustrating the phases of the Moon in relation to the Earth and the Sun. It is a circular diagram with the Sun at the top right and the Earth at the bottom left. The Moon is shown as a crescent in the top left, waxing crescent, first quarter, waxing gibbous, **full, waning gibbous**, third quarter, and waning crescent. Arrows indicate the direction of the waxing and waning of the Moon's phases. The background is solid black, emphasizing the diagram's white and gray elements. There are no background elements or colors other than the white and gray of the Moon and Earth, and the black of the background.
**Prediction:** C. waning Gibbous. ( ✔ )

Table 17: **PrismCaptioner Generates Description to Solve Science & Technology Task.**

**Question: For case A accompanying table, answer the questions that follow. Calculate the future value of the annuity, assuming that it is an ordinary annuity.**
**Image:**

| Case | Amount of annuity | Interest rate | Deposit period (years) |
|------|-------------------|---------------|------------------------|
| A | $ 2,500 | 8% | 10 |
| B | 500 | 12 | 6 |
| C | 30,000 | 20 | 5 |
| D | 11,500 | 9 | 8 |
| E | 6,000 | 14 | 30 |

**Options:** A. $126,827     B. $223,248     C. $36,216     D. $4,058
**Category:** Math
**Description:** The image displays a table with five rows and four columns, excluding the header row. The header row is labeled from left to right as "Case," "Amount of annuity," "Interest rate," and "Deposit period (years)." Below the header, there are five cases labeled A, B, C, D, and E. The "Amount of annuity" column lists monetary values: $2,500 for case A, $500 for case B, $30,000 for case C, $11,500 for case D, and $6,000 for case E. The "Interest rate" column lists percentage values: 8% for case A, 12% for case B, 20% for case C, 9% for case D, and 14% for case E. The "Deposit period (years)" column lists numerical values: 10 years for case A, 6 years for case B, 5 years for case C, 8 years for case D, and 30 years for case E. The table is set against a plain background with no additional elements or decorations.
**Prediction:** To calculate the future value of an ordinary annuity, you can use the formula:
$FV = Pmt * [(1 + r)^n - 1]/r$,
where FV = Future Value, Pmt = Amount of annuity ($2,500), r = Interest rate (8% or 0.08), n = Deposit period (10 years).
Plugging in the values:
$FV = \$2,500 * [(1 + 0.08)^{10} - 1]/0.08$
$FV = \$2,500 * [(1.08)^{10} - 1]/0.08$
$FV = \$2,500 * [2.158924 - 1]/0.08$
$FV = \$2,500 * 1.158924/0.08$
$FV = \$36,216.55$
Therefore, the correct answer is: C. $36,216 ( ✔ )

Table 18: **PrismCaptioner Generates Description to Solve Math Task.**

| Question | Where is the rug in the living room located? |
|---|---|

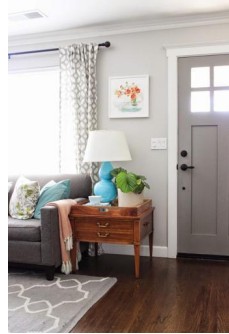

| Options | A. Next to the door | B. Under the table |
|---|---|---|
| | C. In front of the window | D. Under the couch |

| Models | GPT-4o | PrismCaptioner |
|---|---|---|
| **Descriptions** | The image depicts a cozy living room corner with a modern and inviting aesthetic. Here is a detailed description of the scene:
1. **Furniture and Objects:**
Sofa: A gray upholstered sofa is positioned on the left side of the image. It has a few decorative pillows in various patterns and colors, including white, green, and teal.
Side Table: Next to the sofa, there is a wooden side table with two drawers. On top of the table, there is a blue ceramic lamp with a white lampshade and a potted plant with large green leaves.
Rug: A gray rug with a white geometric pattern is placed on the floor. The rug is positioned in front of the sofa and extends partially under it, covering the wooden floor.
Curtains: Light gray curtains with a subtle pattern hang from a black curtain rod above a window on the left side of the image.
Wall Art: Above the side table, there is a framed piece of art featuring a colorful floral arrangement.
**2. Placement and Relationships:**
The rug is centrally placed **in front of the sofa, extending slightly under it**, and covers a portion of the wooden floor.
The side table is positioned between the sofa and the door, with the lamp and plant on top providing both functional and decorative elements.
The framed art is centered above the side table, adding a pop of color to the wall.
...
Overall, the arrangement of furniture and decor items creates a balanced and aesthetically pleasing living room corner. The rug's placement in relation to the sofa and side table helps define the seating area and adds warmth to the space. | The image shows a neatly arranged living room corner with a focus on a side table and a portion of a couch. The side table, which appears to be made of wood with a natural finish, has two drawers with metal handles. On top of the table, there is a large blue lamp with a white lampshade, a small white bowl, and a green potted plant with broad leaves. To the left of the side table, there is a couch with a grey upholstery, adorned with a teal throw pillow and a pink throw blanket casually draped over it. The floor is covered with a grey and white patterned area rug, and the walls are painted in a light grey tone. A window with white trim is partially visible, dressed with patterned curtains in shades of grey and white. On the wall, there is a framed artwork featuring a bouquet of flowers in a vase. The door in the background is a dark grey with a black doorknob, and there is a light switch plate visible on the wall to the right of the door. The overall aesthetic of the room is modern with a touch of warmth provided by the wooden furniture and the cozy textiles. |
| **Predictions** | D. Under the couch ( ✔ ) | C. In front of the window ( 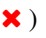 ) |

Table 19: **Performances between GPT-4o and PrismCaptioner.**

| Question | What is the relative position of the man and the woman sitting at the table? |
|---|---|

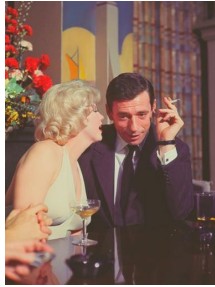

| Options | A. The man is on the left and the woman is on the right |
|---|---|
| | B. The man is on the right and the woman is on the left |
| | C. The man and the woman are facing each other |
| | D. The man and the woman are back-to-back |

| Models | GPT-4o | PrismCaptioner |
|---|---|---|
| Descriptions | The image depicts a man and a woman sitting closely together at a table in what appears to be a social setting, possibly a bar or a restaurant. The woman, **on the left**, has blonde hair styled in loose curls and is wearing a white dress with a plunging neckline. She is leaning in towards the man, appearing to whisper something in his ear. Her left hand is resting on the table, and her right hand is not visible.
The man, **on the right**, is wearing a dark suit with a white shirt and a dark tie. He is holding a cigarette in his right hand, which is raised near his face. His left hand is resting on the table. He is looking directly at the camera with a slight smile.
On the table in front of them, there is a champagne glass near the woman and a glass with a dark liquid, possibly a cocktail, near the man. There are also other objects on the table, including what appears to be a hand of another person partially visible on the left side of the image. In the background, there is a colorful arrangement of red and white flowers on the left side. The backdrop includes a painted mural or artwork featuring abstract shapes and possibly a cityscape with buildings and a sailboat.
The spatial arrangement shows the man and woman sitting very close to each other, with the woman leaning in towards the man, indicating a sense of intimacy or a private conversation. The man is positioned slightly turned towards the camera, while the woman is more focused on him. | The image depicts a man and a woman seated at a table in what appears to be a bar or restaurant setting. The man is dressed in a dark suit with a white shirt and a dark tie, and he is holding a cigarette between his fingers. The woman is wearing a white dress with a plunging neckline, and her blonde hair is styled in curls.
Both individuals are looking at each other, and the woman seems to be whispering into the man's ear. On the table, there are two glasses, one of which appears to contain a clear liquid, possibly water, and the other is not fully visible. There is also a small, dark object that could be an ashtray or a coaster.
In the background, there is a bouquet of flowers with red and white blooms, and the wall behind them features a large, abstract painting with blue and yellow tones. The lighting in the scene is warm, and the overall ambiance suggests a casual yet intimate setting. |

| Predictions | A. The first image ( ✔ ) | B. The second image ( ✘ ) |
|---|---|---|

Table 20: **Performances between GPT-4o and PrismCaptioner.**

