# OpenReview forum: "Prism: A Framework for Decoupling and Assessing the Capabilities of VLMs"
_NeurIPS.cc/2024/Conference — NeurIPS 2024 poster_

### Official Review · Reviewer_onkG · 2024-06-27

**Soundness:** 2
**Presentation:** 3
**Contribution:** 3
**Rating:** 6
**Confidence:** 4

**Summary:**

This paper introduces Prism, a framework to disentangle and evaluate the perception and reasoning capability of vision-language models (VLMs). They evaluate state-of-the-art VLMs, including proprietary ones and open-source ones, with varying model sizes. The evaluation results demonstrate that VLMs' perception ability is consistent regardless of the language encoder size while the reasoning ability is constrained by the model size. Moreover, they develop a vision-language model based on the discovered principles, which is lightweight yet effective.

**Strengths:**

1. The paper proposes a novel framework to evaluate disentangled perception and reasoning abilities of VLMs.
2. The paper conducts thorough experiments to validate the framework and evaluate state-of-the-art VLMs. Various proprietary and open-source VLMs with different model sizes are covered.
3. Based on experimental conclusions, a lightweight yet effective VLM is developed, showing the soundness of the conclusions.

**Weaknesses:**

1. In the perception capability evaluation, besides accuracy, how detailed the image captions given by the VLMs are can also affect the final accuracy. As different VLMs are pre-trained on different datasets for different down-stream purposes, some of them might be trained to give concise response, leading to low final accuracy in Prism. However, giving concise response does not necessarily mean the models cannot see the other details. Therefore, Prism might be inaccurate to evaluate their perception capability.
2. Different VLMs are pre-trained on different datasets, which could cause their differences in perception and reasoning capability. It would provide more insights if the paper investigates the relationship between the models' pre-training datasets and their perception and reasoning capabilities.
3. The results of query-specific instruction in perception ability evaluation is not only affected by the models' capability in perception, but also their capability in instruction-following, e.g., the second example in Figure 3. It would be more accurate to reflect the perception capability by first checking whether the model response follows the instruction, e.g., by using an external LLM.

**Questions:**

1. To evaluate a VLM's perception capability, Prism uses the VLM as an image captioner and feeds the generated caption to an external LLM to answer the question. The final accuracy can indicate how accurately the VLM can describe the image. However, evaluating the VLM on image captioning benchmarks can also reflect the perception ability and is more straightforward than the proposed framework. What are the advantages of Prism over image captioning benchmark evaluation?

**Limitations:**

Limitations and broader impacts are discussed in Appendix D.

---

> ### Author Rebuttal · Authors · 2024-08-07
>
> We thank the reviewer for the valuable feedback. We are uncertain if there is some misunderstanding and would like to clarify that **VLMs** discussed in our paper specifically refer to **large visual language models (LLaVA, GPT-4v, etc.)** designed for **solving general visual language tasks** (as stated in Line 23 and Line 26). We are encouraged that the reviewer believes our work "proposes a novel framework to evaluate disentangled perception and reasoning abilities of VLMs". We address the reviewer's main concerns below.
>
> Q1. **The Evaluation Might Be Inaccurate**
>
> The reviewer suggests that Prism evaluation might be unfair for VLMs that are pre-trained for specific purposes and generate concise responses. However, we would like to clarify that such a premise doesn't align with the assumption of this work. We adopt the Prism framework to evaluate **large visual language models (LLaVA, GPT-4v, etc.)** designed for **solving general visual language tasks**. As stated in Line 27, we define the perception capability as "extracting necessary information from the image", which is beyond "seeing" (we are not sure if a VLM can "see" if it cannot clearly express what it sees). By design, VLMs for general tasks should be able to follow the instructions and express detailed information, regardless of response patterns or training strategies. Thus, we believe that perception evaluation under the Prism framework is a reasonable approximation of true perception capacity.
>
> Q2. **Effects of Pre-training Dataset on VLM's Abilities**
>
> We adopt the LLaVA architecture with InternLM2-7B to investigate the relationship between VLM's pre-training datasets and their perception abilities under the Prism evaluation framework. We construct four different datasets based on ALLaVA dataset to study the impact of descriptive and text-only data:
>
> - **ALLaVA-Caption:** a set of all descriptive data in ALLaVA.
>
> - **ALLaVA-QA:** a set of all the QA data used for instruction fine-tuning in ALLaVA with the same amount and images as ALLaVA-Caption.
>
> - **ALLaVA-Caption-1xText:** a mix of one copy of text-only data in ALLaVA with ALLaVA-Caption, with a ratio of 1:5 (text-only:image-text)
>
> - **ALLaVA-Caption-2xText:** a mix of two copies of text-only data in ALLaVA with ALLaVA-Caption, with a ratio of 2:5 (text-only:image-text)
>
> The performance of their perception abilities on MMStar is shown below.
>
> | Dataset | Perception Performance (Generic) | Perception Performance (Query-Specific) |
> | -- | -- | -- |
> | ALLaVA-Caption        | 40.9  | 43.6 |
> | ALLaVA-QA             | 38.3  | 37.0 |
> | ALLaVA-Caption-1xText | 41.7  | 44.3 |
> | ALLaVA-Caption-2xText | 40.9  | 44.5 |
>
> The results of ALLaVA-Caption and ALLaVA-QA reveal that utilizing the descriptive data better triggers the VLM’s ability to extract and articulate visual information compared with QA data. Focusing on text-only data, one can observe that training with a small amount of text data can improve the VLM's ability to extract and express information. When the ratio of text data increases, VLM shows a degradation in the perception performance of generic instructions, indicating that an appropriate data recipe is crucial for VLM's perception capabilities.
>
> As for the reasoning aspect of VLM, it is difficult for Prism to give a precise metric to quantify it. Prism evaluation framework is more about assessing whether the overall performance of the VLM is limited by its reasoning capabilities. We will further investigate the particular impact of training data on reasoning in subsequent work.
>
> Q3. **Instruction Following Failures**
>
> Excluding all cases of instruction following failures can potentially be a way to assess the true perception capabilities more accurately. However, such practice is rarely adopted. First of all, if a VLM fails to extract the relevant information according to given instructions, it's difficult to know whether the issue lies with the instruction following or the VLM's perception capabilities. Meanwhile, most existing multi-modal benchmarks (MMMU, MathVista, etc.) do not make specific adjustments to mitigate the effect of instruction following.
>
> Q4. **Image Caption Benchmarks**
>
> Directly evaluating the quality of image description can always be the best way to assess the perception capability. However, accurately evaluating image description quality is a significant challenge, and using traditional caption metrics is not a viable solution. Traditional image caption metrics are too sensitive to the caption styles and are not comprehensive. To validate this, we conduct thorough experiments on COCO Caption (val). The partial results are as follows.
>
> | Model | BLEU-1 | BLEU-4 | ROUGE-L | CIDEr |
> | -- | -- | -- | -- | -- |
> | Qwen-VL-Chat | 75.8 | 34 | 54.9 | 98.9 |
> | InstructBLIP-7B | 56.8 | 20.9 | 39.9 | 58.1 |
> | GPT-4o | 21.2 | 3.9 | 20.3 | 0 |
> | GPT-4v | 18 | 3.3 | 18.1 | 0 |
> | InternVL-Chat-v1.5 | 15.9 | 3 | 15.8 | 0 |
> | LLaVA-Next-Yi-34B  | 12.8 | 2.4 | 13.1 | 0 |
>
> Excellent VLMs, such as GPT-4v, GPT-4o, and InternVL-Chat-v1.5, struggle to score well, with all metrics significantly lower than the low-profile Qwen-VL-Chat (the reason is that those advanced VLMs generate much longer responses compared to the ground-truth caption). This contradicts the consensus and further indicates that the caption benchmarks are unsuitable for assessing VLMs' perception abilities. In Prism, we use an external LLM to answer the question based on the image description generated by VLMs to evaluate the VLM's perception capability. Compared to the caption evaluation, the score obtained with Prism is more comprehensive and less sensitive since whether an LLM can successfully answer the question is basically determined by the quality of the descriptive text, not the style. Moreover, existing image caption benchmarks are often limited to specific domains. In Contrast, Prism can be applied to any general multi-modal benchmark to study the VLM's capability.

---

> > ### Comment · Reviewer_onkG · 2024-08-11
> >
> > I read the other reviewers' review and the authors' responses to them. I revisited the model prompts and understood that they are instructed to give very detailed image descriptions. And I agree with the authors that failures in instruction following is also a type of perception failure. I would advise the authors to discuss the proportion of the different types of failures mentioned in their response to reviewer KmBE. Moreover, I highly appreciate the authors' discussion about Prism's advantages over image captioning evaluation. Therefore, I raise my rating to 6 (Weak Accept).

---

> ### Author Response · Authors · 2024-08-10
>
> Dear Reviewer onkG,
>
> We greatly appreciate the time and effort you've taken to review our submission. We hope that our response has addressed the concerns raised in your initial reviews, and we look forward to your feedback.
>
> As the author-reviewer discussion period for NeurIPS 2024 is half past, please let us know if you need any further information or clarification. We are fully open to engaging in further discussions to improve our work.
>
> Best regards and thanks,
>
> Paper 8561 Authors

---

> ### Author Response · Authors · 2024-08-13
>
> We thank the reviewer for the encouraging comments. We are pleased to note that our response could address some of the reviewer's concerns.
>
> Regarding the proportion of the different types of failures, we sample 100 cases from all VLMs and conduct quantitative analysis based on the categorization in our response to the reviewer KmBE. There are 72 perception failures and 34 reasoning failures in all 100 cases. In some cases, both perception and reasoning errors exist. The detailed proportion result of errors (in perception/reasoning) is as follows:
>
> - **Perception:**  **Factual Errors (43.1%)**, **Incomplete Details (40.2 %)**, **Instruction Following (16.7%)**
> - **Reasoning:**  **Logical Errors (67.6%)**, **Lack of Knowledge (26.5%)**, **Misunderstanding of the Query (5.9%)**
>
>
>
> For the cases where perception and reasoning errors both exist, here is an example from MMStar:
>
> Index: 1222
>
> Image: A bar chart titled "Accuracy of different algorithms". The vertical axis is labeled "Accuracy" and ranges from 0 to 10. The horizontal axis has two categories: "ivory" and "calf." The height of "ivory" is 9, and that of "calf" is 4.
>
> Question:  What is the sum of the accuracies of the algorithms calf and ivory?
>
> LLaVA-NeXT (Vicuna-13B) succeeds in obtaining the heights of two categories but gives the wrong horizontal interval, 0-9, which is a perception error. ChatGPT gives the wrong sum, 11, even if the heights 4 and 9 are correctly expressed by the VLM, which is a reasoning error.
>
>
>
> Once again, we thank the reviewer for the time and insights. We hope this comment could address the reviewer's concerns and will make sure to properly incorporate the additional discussions in the rebuttal into our revised paper.

---

### Official Review · Reviewer_rB4V · 2024-07-12

**Soundness:** 3
**Presentation:** 3
**Contribution:** 3
**Rating:** 7
**Confidence:** 4

**Summary:**

The paper introduces Prism, a framework designed to decouple and independently assess the perception and reasoning capabilities of VLMs. Prism operates in two stages: a perception stage that extracts visual information and converts it into text using a VLM, and a reasoning stage that generates answers based on this textual information using a LLM. This modular approach allows for systematic evaluation of each component. Several insightful results are presented through extensive experiments.

**Strengths:**

* the paper is well-organized and easy to follow
* the experiments are extensive and the analysis provides some valuable insights
* the Prism is effective and demonstrates competitive performance on several benchmarks
* decoupling the end-to-end inference into perception and reasoning stages bring a new approach to solve the tasks that require complex reasoning process

**Weaknesses:**

* The prompt design for the perception and the reasoning stage is an important aspect. The impact of different prompt designs should be analyzed.
* Decoupling the inference into two stages might introduce additional computational overhead and latency, and cost-effectiveness needs to be discussed.
* Prism relies on language descriptions; however, in some scenarios, language struggles to describe the content of images because certain visual or logical concepts lack corresponding linguistic expressions, e.g. medical pathology images and graphs of mathematical functions.

**Questions:**

* Another line of work focuses on assessing the compositional or fine-grained understanding abilities of VLMs, such as [1][2][3][4]. Could you discuss the relevance or differences of these works to Prism?
* Can you discuss the advantages and disadvantages of the 'end-to-end' approach versus the 'decoupling' approach?


My overall judgment of this article is positive. I am open to raising my score if the author can address my concerns listed above.

****
[1] Winoground: Probing Vision and Language Models for Visio-Linguistic Compositionality, CVPR 2022

[2] Synthesize, Diagnose, and Optimize: Towards Fine-Grained Vision-Language Understanding, CVPR 2024

[3] Diagnosing the Compositional Knowledge of Vision Language Models from a Game-Theoretic View, ICML 2024

[4] CounterCurate: Enhancing Physical and Semantic Visio-Linguistic Compositional Reasoning via Counterfactual Examples, ACL 2024

**Limitations:**

Maybe the authors need to discuss potential limitations and future directions for improvement.

---

> ### Author Rebuttal · Authors · 2024-08-06
>
> We thank the reviewer for the encouraging comments and address the main concerns below. Limitations and broader impacts have been discussed in Appendix. D. We are inspired that the reviewer believes our work "provides some valuable insights" and "brings a new approach to solve the tasks that require complex reasoning process." These are indeed what we want to emphasize.
>
> Q1. **Prompt Design**
>
> We carefully designed the prompts used in the Prism framework:
>
> 1. **Generic Instruction (Perception):** The generic instruction provides common perspectives when describing images. We refer to MathVista and consider scenes, objects, relationships, and text. We further add instance location into the instruction and obtain better results. We experiment with variants of generic instructions and choose the relatively better one, as shown in Table 2.
> 2. **Query-Specific Instruction (Perception):** We incrementally construct the query-specific instruction by merging the generic instruction with the query-specific part. Generating the query-specific part is a reasoning task assigned to LLM. Meanwhile, it is essential to ensure that only the `contents to observe` appear in the instruction since redundant content (questions, etc.) can stimulate the reasoning capability of VLMs. We design precise tasks and add in-context examples for the LLM, as shown in Fig. 7.
> 3. **Instruction for Reasoning:** Visual descriptions and questions can be lengthy and complex, which easily leads to confusion for LLMs. We prompt the LLM to act like a text-based reasoning expert to derive the answer based on the provided information, which shows the best results during our iterations. The prompt template is shown in Fig. 8.
>
> Q2. **Computational Cost and Latency**
>
> 1. **Training Cost**: For a large end-to-end VLM (like VLM based on Llama3-70B), the training cost is extremely high regarding hardware resources and time. Prism offers an alternative which only requires training a lightweight VLM as the perception module, significantly reducing the VRAM usage (GPUs with 40/80GB VRAM are still scarce and expensive) and the training computation. Prism doesn't tune the LLM. The total training cost of Prism is much less than that of an end-to-end VLM equipped with large language encoders (like Llama3-70B).
> 2. **Deployment Cost**: The user spends much less cost to deploy a lightweight visual captioner compared to a large-scale VLM. On the LLM side, thanks to the advanced deployment techniques of LLMs, the LLM inference APIs are available at an extremely low price (also see Appendix D), so the user can take advantage of vast numbers of LLM APIs with low financial cost.
> 3. **Latency**: Prism requires a VLM to generate intermediate descriptive text, bringing additional computation overhead. When solving a single visual question (VQ), a higher latency is expected compared to the end-to-end VLM with a language encoder of the same size. However, when one asks for multiple VQs corresponding to one image, Prism will display a smaller latency since the descriptive text can be reused. Moreover, the advanced LLM deployment techniques (compared to VLMs) further help reduce Prism's latency.
>
> Q3. **Lack of Linguistic Expressions for Some Images**
>
> In Appendix. D, we mentioned that the current PrismCaptioner (learned on the general multi-modal corpora ALLaVA) may struggle to describe images in unseen domains (GUIs, medical images, etc.).   However, we do believe that almost all images can be described in detail by linguistic expressions. For example, a well-trained doctor can describe a medical image in detail with natural language (only noteworthy things, not all pixels, need to be described) and a graduate student can convert a screenshot of math equations into natural language descriptions or latex. With sufficient high-quality visual instruction tuning data, PrismCaptioner also has the potential to master these professional tasks.
>
> Q4. **Relevance and Differences of Referred Works to Prism**
>
> We would incorporate those literature in related works and include a detailed discussion. The relevance and differences between Prism and these works lie in the following aspects:
>
> 1. **Motivation**: Prism and referred works have different motivations: referred works mainly focus on evaluating visio-linguistic compositional knowledge or fine-grained understanding (mostly perception tasks), while Prism focuses on analyzing the perception and reasoning abilities of VLMs in a decoupled manner.
> 2. **Methodology:** Prism and referred works all establish an evaluation system to analyze the capabilities of interest. [1, 2, 4] construct benchmarks to evaluate certain capabilities, [3] establish a new paradigm to study the compositional knowledge of VLMs, while Prism creates a decoupling framework to study the perception and reasoning capabilities, respectively.
> 3. **Application:** From the application perspective, Prism can not only serve as an evaluation framework but also effectively address visual language tasks.
>
> Q5. **Decoupling Versus End-to-End (e2e)**
>
> 1. **Cost and Latency**: As discussed in Q2, to reach similar performance, the decoupling paradigm features a smaller training and deployment cost (VRAM usage, hardware resources, financial costs, etc.) compared to the e2e approach. For a single visual question, Prism shows a higher latency compared to the e2e VLM. Meanwhile, the average latency of Prism can be lower when Prism deals with a batch of VQs corresponding to the same image.
> 2. **Performance**:  With abundant computation resources and training data to train a large-scale VLM (equipped with large language encoders such as Llama3-70B), adopting the decoupling paradigm may not be a good choice for the final performance. However, under resource-limited (computation, training data, financial budget, etc.) scenarios, Prism can outperform e2e VLMs regarding performance and flexibility (please refer to Tables. 5, 6).

---

> ### Author Response · Authors · 2024-08-10
>
> Dear Reviewer rB4V,
>
> Thank you for the time and effort you have dedicated to reviewing our submission. We hope our response has effectively addressed the concerns raised in your initial reviews, and we eagerly await your thoughts and further guidance to refine our work.
>
> As the author-reviewer discussion period for NeurIPS 2024 is half past, please let us know if you require any additional information or clarification. We are more than willing to engage in further discussions to enhance our work.
>
> Best regards and thanks,
>
> Paper 8561 Authors

---

> > ### Comment · Reviewer_rB4V · 2024-08-12
> >
> > I appreciate the response from the authors. Most of my concerns have been resolved. However, regarding Q3, I still believe that some information is difficult to accurately describe at the linguistic level and instead needs to be represented in a more abstract space. I hope the author can provide a more in-depth discussion on this.
> > Overall, I am inclined to accept this paper, and I will raise my score to 7.

---

> ### Author Response · Authors · 2024-08-13
>
> We express our gratitude to the reviewer for the constructive feedback. We are encouraged to learn that our response could address some of the reviewer's concerns.
>
>
>
> In terms of Q3: **Lack of Linguistic Expressions for Some Images**, we believe most images can be described in detail by linguistic expressions, even if they are medical pathology images or math contents. However, we acknowledge that natural languages may struggle to express some obscure visual elements, especially some aesthetic contents, e.g., abstract artworks and surrealist paintings. In these cases, it is difficult to obtain high-quality descriptions. Thus, we will further focus on potential representation in a more abstract way instead of linguistic expressions.
>
>
>
> Once again, we thank the reviewer for the time and effort. If our response could address the reviewer's concerns, we hope the reviewer could raise the score. (now it is 6.) We will ensure that the additional discussions in the rebuttal are properly incorporated into our revised paper.

---

### Official Review · Reviewer_KmBE · 2024-07-13

**Soundness:** 4
**Presentation:** 3
**Contribution:** 3
**Rating:** 7
**Confidence:** 4

**Summary:**

In this paper, the authors propose prism, a framework to decouple the VLMs' capabilities in two stages: perception stages and reasoning stages. This framework allows the breakdown analysis of VLM capabilities and can also serve as a framework to integrate any given VLM and LLM. Based on their explorations and decoupled analysis, they discover that the integration of a lightweight VLM combined with a powerful LLM can be useful and exhibit outstanding performance and efficiency. The author provide good amount of experimentations to support their claims.

**Strengths:**

1. The author present good analysis, findings and insights based on their framework. The insights are valuable.
2. The author provide decent amount of experimentation results on many VLMs. The author demonstrate the soundness and effectiveness of their framework with experimentations.
3. Prism can be useful in both evaluation and task solver.

**Weaknesses:**

1. There is not much unique and novel contributions in terms of algorithms and model designs.

**Questions:**

See weakness section.

1. How about failure modes in different part of the VLMs (reasoning and perception)? Is there any analysis or ideas on analyze the hallucinations based on your framework?

**Limitations:**

limitations and broader impacts are properly addressed in the appendix.

---

> ### Author Rebuttal · Authors · 2024-08-07
>
> We express our sincere gratitude to the reviewer for the constructive feedback. We are glad that the reviewer appreciates the "good analysis, findings, and insights" presented in this work and recognizes the potential of the Prism framework in both evaluation and as a task solver. We address the concerns of the reviewer below:
>
> Q1. **Lacks Novel Contributions in terms of algorithms and model design:**
>
> In this work, we aim at designing **a straightforward framework to disentangle and analyze the perception and reasoning capability** of VLMs. In pipeline design, we come up with the **query-agnostic and query-specific settings** to present more comprehensive evaluation results. When developing the PrismCaptioner, we also aim at **providing the simplest possible baseline** to validate the hypothesis that Prism equipped with a small-scale VLM and a large-scale LLM can potentially be **a powerful multi-modal task solver**. Thus, we select the most widely adopted LLaVA architecture and simply adopt ALLaVA as the instruction tuning corpora. Developing more sophisticated tuning algorithms or more advanced model architecture would improve the Prism framework's overall performance, but it's not our focus, and we leave it to future works.
>
> Q2. **Error Mode and Hallucination**
>
> We conduct a thorough analysis and categorize errors in perception and reasoning into the following major modes (the content will be added to the refined version of this manuscript):
>
> 1. **Major Error Modes (Perception):**
>    - **Factual Errors:** VLMs may describe images with inaccuracies, such as stating that prominent elements are "not visible".
>    - **Incomplete Details:** Even in the absence of factual errors, VLMs may lack detailed content, resulting in insufficient information for reasoning.
>    - **Instruction Following:** VLMs sometimes fail to follow instructions when providing corresponding descriptions.
> 2. **Major Error Modes (Reasoning):**
>    - **Logical Errors:** LLMs may produce incorrect conclusions or reasoning processes due to limited reasoning abilities.
>    - **Lack of Knowledge:** The absence of relevant domain knowledge prevents LLM from solving corresponding problems, especially in specific fields.
>    - **Misunderstanding of the Query**: In rare cases, the query-specific part generated by LLMs deviates from the original question, misleading the perception of VLMs.
>
> Regarding the issue of hallucinations, identifying their source is crucial in our cascading framework. We can leverage various powerful LLMs to reason about the descriptions generated by a VLM and analyze the patterns in their reasoning results. Here are three potential scenarios.
>
> 1. If most LLMs yield the correct answer while one LLM yields an incorrect one, the latter may struggle with reasoning or experience a hallucination.
> 2. If most LLMs indicate a wrong answer, it likely means that hallucinations generated by the VLM cause some misdirection.
> 3. If the answers of the LLMs are chaotic or if there are refusals to answer, it may indicate that the VLM did not provide sufficient detailed information.
>
> We conduct experiments with various VLMs, by selecting a set of powerful language models, including GPT-3.5-Turbo-0125, GPT-4-Turbo-0125, Llama-3-70B-Instruct, and DeepSeek-v2-Chat. For each question, we gather results from four LLMs and consider cases where the same choice appears three times or more as "agreement". In agreement cases, we define the corresponding choice as "voted choice" and focus on the following cases:
>
> - Case 1: With descriptive texts generated by a specific VLM, the voted choice is wrong.
> - Case 2: With descriptive texts generated by a specific VLM, the voted choice is correct.
> - Case 3: For a specific LLM, it makes the same choice as the voted choice by all LLMs, while the voted choice is also the correct one.
>
> For each VLM, we analyze the rate of case 1, which indicates the potential hallucinations of VLMs. For each LLM, we correspondingly calculate the rate of case 3 to case 2 to observe the alignment of its predictions with voted choices. The higher alignment rate means more robust reasoning since voted options are considered better. The notions and results are as follows.
>
> $$
> \text{Agreement Rate (VLM)} = \frac{\text{number of agreement cases}}{\text{number of all cases}}
> $$
>
> $$
> \text{Error Rate (VLM)}=\frac{\text{number of case 1}}{\text{number of agreement cases}}
> $$
>
> $$
> \text{Alignment Rate (LLM)}=\frac{\text{number of case 3}}{\text{number of case 2}}
> $$
>
> | VLM | Agreement Rate (VLM) | Error Rate (VLM) |
> | -- | -- | -- |
> | GPT-4o | 68.6 | 29.9 |
> | GPT-4v | 61.5 | 38.0 |
> | LLaVA-NeXT (Yi-34B) | 60.7 | 43.4 |
> | LLaVA-v1.5-7B | 50.6 | 56.5 |
>
> | LLM                  | Alignment Rate (LLM) |
> | -------------------- | ---- |
> | GPT-3.5-Turbo-0125   | 87.1 |
> | DeepSeek-v2-Chat     | 91.3 |
> | Llama-3-70B-Instruct | 92.8 |
> | GPT-4-Turbo-0125     | 90.7 |
>
> The results show that stronger VLMs show lower error rates, indicating that more capable VLMs experience fewer hallucination issues. By delving deep into the specific cases, we find that VLMs are prone to hallucinations in spatial awareness and fine-grained perception.
>
> All LLMs show relatively good alignment rates, demonstrating relatively robust reasoning performance. GPT-3.5-Turbo-0125 is more unstable by comparison. The cases where LLM's prediction does not align with the voted choice may stem both from reasoning ability issues and hallucinations. It requires further manual checking and labeling to clarify whether the failure in alignment is caused by LLM's hallucination.

---

> ### Author Response · Authors · 2024-08-10
>
> Dear Reviewer KmBE,
>
> Thank you for the time and patience you have dedicated to reviewing our submission. We hope we have addressed the concerns raised in your initial reviews and eagerly await your thoughts and further guidance to refine our work.
>
> As the author-reviewer discussion period for NeurIPS 2024 is half past, please let us know if you require additional information or clarification. We are eager and ready to engage in further discussions to enhance and elevate our work.
>
> Best regards and thanks,
>
> Paper 8561 Authors

---

### Decision · Program_Chairs · 2024-09-25

**Decision:**

Accept (poster)

**Comment:**

Systematic analyzing VLMs is still a challenging and open research topic, the authors presented an aspect of doing it through decoupling framework. All reviewers and AC hold positive view of this work, and suggest accept.